# Enhancement of single upconversion nanoparticle imaging by topologically segregated core-shell structure with inward energy migration

Yanxin Zhang[1,3], Rongrong Wen[1,3], Jialing Hu[1], Daoming Guan[1], Xiaochen Qiu[1], Yunxiang Zhang [1]✉, Daniel S. Kohane [2]✉ & Qian Liu [1]✉

Manipulating topological arrangement is a powerful tool for tuning energy migration in natural photosynthetic proteins and artificial polymers. Here, we report an inorganic optical nanosystem composed of $NaErF_4$ and $NaYbF_4$, in which topological arrangement enhanced upconversion luminescence. Three architectures are designed for considerations pertaining to energy migration and energy transfer within nanoparticles: outside-in, inside-out, and local energy transfer. The outside-in architecture produces the maximum upconversion luminescence, around 6-times brighter than that of the inside-out at the single-particle level. Monte Carlo simulation suggests a topology-dependent energy migration favoring the upconversion luminescence of outside-in structure. The optimized outside-in structure shows more than an order of magnitude enhancement of upconversion brightness compared to the conventional core-shell structure at the single-particle level and is used for long-term single-particle tracking in living cells. Our findings enable rational nanoprobe engineering for single-molecule imaging and also reveal counter-intuitive relationships between upconversion nanoparticle structure and optical properties.

Energy migration is an essential process in numerous systems, such as natural photosynthetic proteins, artificial polymers, and inorganic optical materials[1-3]. In particular, in lanthanide-doped upconversion, which absorbs two or more near-infrared photons and emits one visible photon through a multiphoton process, energy migration between sensitizer $Yb^{3+}$ ions show great potential for tailoring luminescence brightness, lifetime, and wavelength[4-7]. Tuning the topological structure of materials has been demonstrated to affect energy migration[8-10]. Based on this, the allocation of lanthanide active ions can have a profound effect on energy migration and luminescence brightness in a core-shell structure, which can be viewed as two interfacing compartments with distinct topologies. The energy migration pattern manifested in the hopping of energy quanta amongst lattice sites occupied by lanthanide ions depends on the topology for the lattice network of sensitizers ions as well as the spatial distribution of active ions, therefore, deserves close investigation.

Lanthanide-doped upconversion nanoparticles (UCNPs) are an important family of optical materials that demonstrate significant advantages over other luminescent probes, including excellent photostability, non-blinking, sharp emission lines, and large anti-Stokes shifts[11-13]. Upconversion luminescence (UCL) offers many applications, such as bioimaging, multicolor displays, super-resolution nanoscopy,

[1]Department of Chemistry and Shanghai Key Laboratory of Molecular Catalysis and Innovative Materials, Fudan University, Shanghai 200438, China. [2]Laboratory for Biomaterials and Drug Delivery, Division of Critical Care Medicine, Children's Hospital Boston, Harvard Medical School, 300 Longwood Avenue, Boston, MA 02115, USA. [3]These authors contributed equally: Yanxin Zhang, Rongrong Wen. ✉e-mail: zyx@fudan.edu.cn; daniel.kohane@childrens.harvard.edu; qianliu@fudan.edu.cn

**Fig. 1 | Schematic diagrams of designs. a** Schematic diagrams of outside-in, inside-out and local energy transfer architectures. The straight black arrows represent the principle energy migration direction of outside-in and inside-out architecture with distinct interface. Orange ball represents $Er^{3+}$, blue ball $Yb^{3+}$, gray ball $Lu^{3+}$; black curved arrows, purple curved arrows and red curved arrows depicts the energy migration, energy transfer and back energy transfer, respectively. **b** Schematic diagrams of energy transfer from $Yb^{3+}$ (sensitizer) to $Er^{3+}$ (emitter) and upconversion emission luminescence. Black curved arrows, purple curved arrows, green arrows, red solid arrows and red dashed arrows depicts the energy migration (EM), energy transfer (ET), green emission, red emission and photon energy absorption respectively. The energy flux of upconversion process included $Yb^{3+}$ ions (sensitizer) absorbed excitation photons, energy migration between $Yb^{3+}$ ions, energy transfer from $Yb^{3+}$ to $Er^{3+}$ (emitter), then the energy was released by emitting a short-wavelength photon.

and photovoltaics[14–19]. In addition, in biological contexts upconversion imaging can eliminate interference from autofluorescence and allows background-free detection[20]. These advantages make UCNPs promising single-molecule imaging probes, which typically suffer from limited photostability and tissue autofluorescence. UCNPs doped with the lanthanide ion $Er^{3+}$ or $Tm^{3+}$ as emitter and $Yb^{3+}$ as sensitizer have been explored as single-molecule imaging probes[21–24]. However, the faint brightness of single nanoparticles limits their application in bioimaging, especially for sub-20 nm UCNPs[25,26]. Smaller nanoparticles significantly decay in brightness due to increased surface quenching and reduced sensitizer and emitter ion numbers per particle[27–30]. Various strategies, such as adding an outmost inert shell, optimizing the doping concentration of active ions, and introducing organic molecules to enhance light harvest ability or energy converted efficiency, have been developed to optimize the upconversion[26,31–34]. However, most of them are based on ensemble measurement. These developed strategies based on ensemble measurement were inspiring but cannot be translated to single-particle's brightness optimization directly. And in most cases, these two are contradictory[35].

To understand the relationship between topological arrangement and energy migration and to systematically optimize the upconversion brightness of sub-20 nm UCNPs at the single-particle level, we designed and prepared a series of ~17 nm core-interior shell-inert shell UCNPs with inside-out (sensitizer@emitter@inert-shell, Fig. 1a), outside-in (emitter@sensitizer@inert-shell), and local energy transfer architecture (sensitizer&emitter@inert-shell). $Yb^{3+}$ was selected as the sensitizer and $Er^{3+}$ as the emitter (Fig. 1b). They were structured so that the $Er^{3+}$ was in the spherical core and $Yb^{3+}$ in the shell (outside-in) or vice versa (inside-out). The core and shell were equal in volume such that effects of the ions in the two compartments could be compared. In a third architecture, $Yb^{3+}$ and $Er^{3+}$ were intermixed in a single core (local energy transfer). The terms "outside-in" and "inside-out" refer to the directions of energy transfer, from sensitizer to emitter. An outmost inert shell of $NaLuF_4$ was used to minimize the surface quenching and improve upconversion efficiency[33] for all three designs. Luminescence spectra and lifetimes were measured in ensemble (i.e., the collective properties of a suspension of nanoparticles) and compared for all three designs. However, comparing such samples can be challenging due to difficulties determining nanoparticle number per unit volume accurately, and aggregations may form over time[35]. To avoid those difficulties, we also performed single-particle characterizations.

Our mechanistic investigation revealed topology-dependent energy migration where energy migrated to the interface more effectively in the outside-in architecture. Therefore, the local density of excitation energy is maximized compared to the other two architectures. When the excitation energy is transferred to emitter $Er^{3+}$, an enhanced luminescence of outside-in UCNPs is obtained under near-infrared irradiation and characterized at the single-particle level. Such single-particle studies of topology-dependent energy migration not only enable the engineering of smaller and brighter single-particle imaging probes but also provide an in-depth understanding of the relationships between structural designs and optical properties.

## Results

### Synthesis and characterization

As proof of concept, we fabricated three upconversion nanoparticles in the above-mentioned topological arrangements of energy transfer

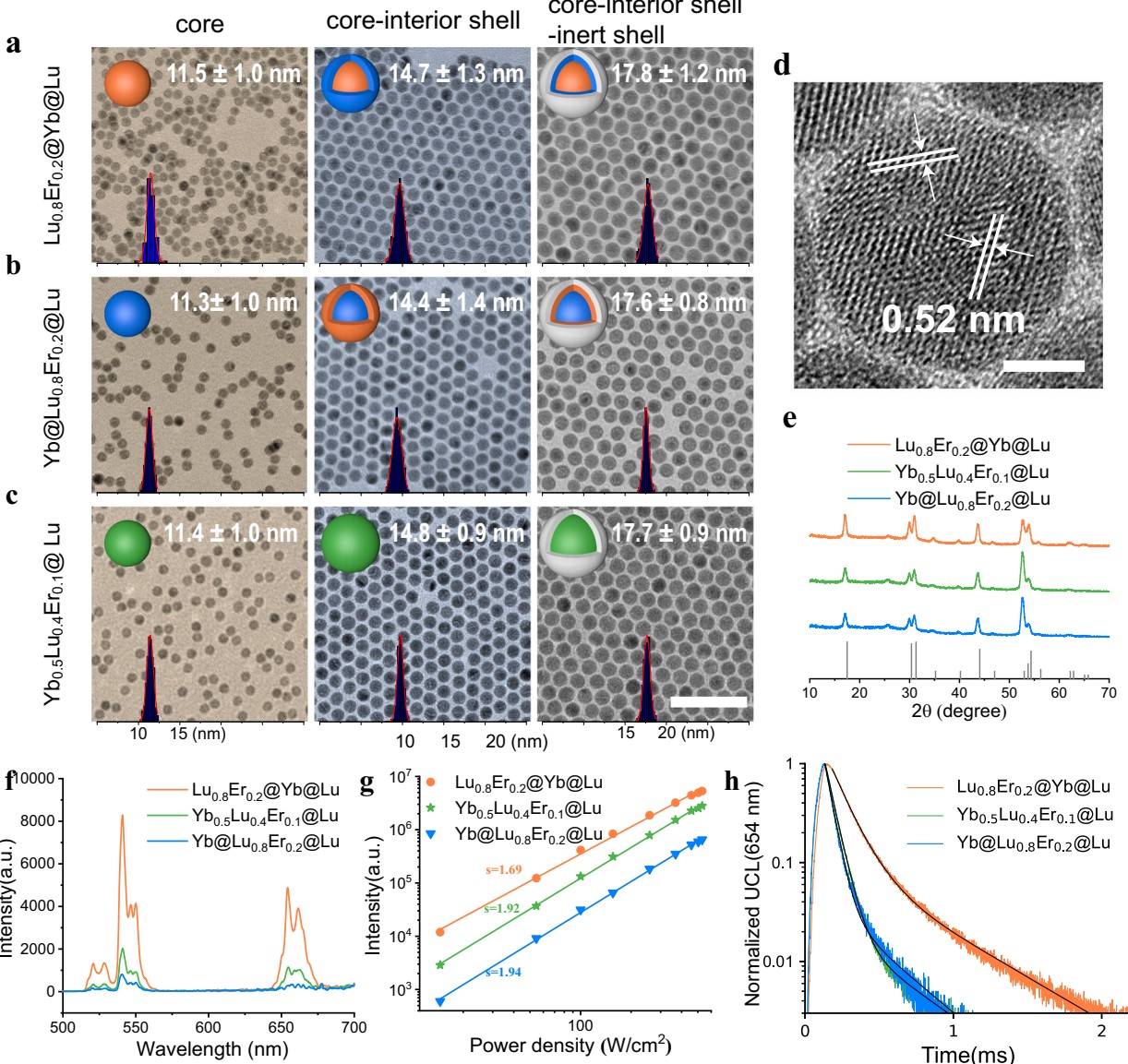

**Fig. 2 | Characterizations of outside-in, inside-out, and local energy transfer core-interior shell-inert shell UCNPs. a–c** Representative TEM images of $Lu_{0.8}Er_{0.2}@Yb@Lu$, $Yb@Lu_{0.8}Er_{0.2}@Lu$ and $Yb_{0.5}Lu_{0.4}Er_{0.1}@$ Lu. Shown from the left to right are the core, the core-interior-shell and the core-interior-shell-inert shell architecture with histograms of size distribution in blue and Gaussian fitting in red overlayed at the bottom of each TEM image. More than five TEM images of each sample were included for statistical analysis, the results were presented as mean ± standard deviation, all TEM images share a scale bar of 100 nm. **d** HR-TEM image of $Lu_{0.8}Er_{0.2}@Yb@Lu$, the lattice spacing is 0.52 nm, corresponding to the (1010) plane of β-$NaLuF_4$. Scare bar, 5 nm. **e** XRD patterns of these core-interior shell-inert shell samples, compared with the standard spectrum of β-$NaLuF_4$ (#27-0726). **f** UCL spectra of core-interior shell-inert shell UCNPs in ensemble solution under 7.3 W/cm² 980 nm laser irradiation. **g** Power-dependent properties of ensemble UCL green emission at 654 nm presented with linear lines of best fit and the associated slopes. **h** Luminescence decay curves, excited by 980 nm pulsed laser and recorded at 654 nm emission. The black lines denote the corresponding double exponential decay fitting. Source data are available as Source Data file.

architectures (Fig. 1): inside-out ($NaYbF_4@NaLu_{0.8}Er_{0.2}F_4@NaLuF_4$ simplified as $Yb@Lu_{0.8}Er_{0.2}@Lu$), outside-in ($NaLu_{0.8}Er_{0.2}F_4@$-$NaYbF_4@NaLuF_4$ simplified as $Lu_{0.8}Er_{0.2}@Yb@Lu$) and local energy transfer architectures ($NaYb_{0.5}Lu_{0.4}Er_{0.1}F_4@NaLuF_4$ simplified as $Yb_{0.5}Lu_{0.4}Er_{0.1}@Lu$). The inside-out and outside-in energy transfer architecture have nearly identical interfaces between sensitizer ions and emitter ions (Fig. 1). In the local energy transfer architecture the active ions are intermixed without an interface. The upconversion process consists of $Yb^{3+}$ absorbing near-infrared photons and migrating energy to nearby $Yb^{3+}$ and then transferring energy to $Er^{3+}$, emitting visible photons.

Unequal numbers of sensitizer and emitter ions in each nanoparticle could affect the resulting upconversion brightness[36]. To

explore the effect of topological arrangement on upconversion brightness without the confounding effect of differences in ion number, we made all the upconversion nanoparticles with the same outer diameter and the same thickness of the inert outer shell (Fig. 2a–c). To minimize the concentration-dependent quenching effect of cross-relaxation between $Er^{3+}$ ions[37], a doping concentration of 20% $Er^{3+}$ was chosen for the emitter layer of the nanoparticles[38]; the remainder of the emitter layer was $Lu^{3+}$. All the nanoparticles were prepared by an epitaxial core/shell growth process as described[25] with varied lanthanide ion doping (see details in methods). The size and morphology of these nanoparticles were characterized by transmission electron microscopy (TEM; Fig. 2a–c), which revealed a uniform size and monodispersity for all. The volume ratio of the core to the interior shell

was -1.0, so that outside-in $Lu_{0.8}Er_{0.2}@Yb@Lu$ and inside-out $Yb@Lu_{0.8}Er_{0.2}@Lu$ UCNPs would contain the same number of sensitizers and emitter ions. Therefore, differences in the numbers of $Yb^{3+}$ and $Er^{3+}$ ions in these nanoparticles would have a negligible impact on upconversion luminescence. Inductively coupled plasma atomic emission spectroscopy (ICP) analysis (Supplementary Table 1) also showed that these UCNPs had almost identical chemical content but differed in topological arrangements. With high-resolution transmission electron microscopy (HR-TEM) and X-ray diffraction (XRD) measurements (Fig. 2d, e), these UCNPs were determined to have a conventional hexagonal phase.

## Ensemble assessment of upconversion properties
We investigated the effect of topological arrangement on ensemble upconversion properties of nanoparticles. The measurement conditions for ensemble characterizations are the same for all the samples such that the results can be compared across various samples. As shown in Fig. 2f, luminescence spectra revealed that all of these three architectures showed significant UCL emissions with peaks at 521, 541, and 654 nm, corresponding to the radiative transitions of $Er^{3+}$ $^4I_{11/2}$, $^4S_{3/2}$, and $^4F_{9/2}$ state, respectively. The outside-in $Lu_{0.8}Er_{0.2}@Yb@Lu$ exhibited enhanced UCL over that of local $Yb_{0.5}Lu_{0.4}Er_{0.1}@Lu$, which suggested that the interfacial energy transfer can improve upconversion luminescence effectively[38–40]. However, the enhancement diminished with increasing irradiation power density (Supplementary Fig. 4 and Fig. 2g). Unexpectedly, inside-out $Yb@Lu_{0.8}Er_{0.2}@Lu$ had a much dimmer luminescence compared with either outside-in $Lu_{0.8}Er_{0.2}@Yb@Lu$ or local $Yb_{0.5}Lu_{0.4}Er_{0.1}@Lu$, even though they had similar architectures and compositions of ions, the principal difference being distinct topological arrangements of the sensitizers and activators, with opposite energy migration directions. It is widely recognized that the UCL intensity has a power-law of index n with respect to the excitation power, where n effectively represents the number of photons involved in upconversion luminescence[41]. Both inside-out and local energy transfer architectures exhibited a typical two-photon upconversion (n-2.0). However, for the outside-in $Lu_{0.8}Er_{0.2}@Yb@Lu$, it showed an enhanced UCL with a shallower slope (n-1.7) in the power dependence curve. We speculate that there was a relatively efficient energy transfer process in the outside-in structure when compared to the inside-out, and the segregation of $Er^{3+}$ and $Yb^{3+}$ that reduces back energy transfer, both of which lead to an enriched $Er^{3+}$ excited intermediates in comparison to the inside-out $Yb@Lu_{0.8}Er_{0.2}@Lu$ and local energy transfer $Yb_{0.5}Lu_{0.4}Er_{0.1}@Lu$. Therefore, the outside-in UCNPs is less sensitive to the power density change than the other two structures.

In order to further understand structure-UCL intensity correlations, we performed lifetime measurements of $Er^{3+}$ ion and found $\tau_{outside-in} > \tau_{local}$ or $\tau_{inside-out}$ (see Fig. 2h, Supplementary Fig. 4 and Table 2; $\tau$, the lifetime, is defined as the time for the intensity to drop by 1/e), which is in accord with the luminescence intensities of the UCNPs. The longer lifetime of outside-in than the local energy transfer architecture could be attributed to the presence of an interface, which minimizes the quenching effect of back energy transfer from $Er^{3+}$ to $Yb^{3+}$. Compared to the outside-in architecture, the decreased lifetime of inside-out may stem from the localization of $Er^{3+}$ ions in the interior shell being closer to the surface quenchers. Similar correlations between luminescence and lifetime have been previously reported for various lanthanide-based UCNPs systems in which a longer luminescent lifetime usually indicates stronger emission intensity[42–45]. In brief, outside-in UCNPs were the most bright architecture for UCL in the ensemble measurement.

## Colocalization and characterization at the single-particle level
As discussed above, single-particle characterizations of upconversion intensity were performed[46,47]. Initially, through the colocalization of

single-particle UCL images and corresponding scanning electron microscope (SEM) images (Fig. 3a, b), we did 1st degree 2D polynomial mapping[48] (see details in Supplementary Methods) and showed a near perfect match between nanoparticles in the wide field luminescence image and those in the SEM image with registration errors as small as half the size of a single nanoparticle, which demonstrated that the signal in UCL images was from individual particles, not clusters. The single-particle brightness under continuous wave wide-field illumination of 980 nm was measured at various excitation power densities from 126 W/cm² to 21.7 kW/cm² (Fig. 3c, d). Similar to the ensemble results, outside-in $Lu_{0.8}Er_{0.2}@Yb@Lu$ exhibited the brightest emission among these three UCNPs, with an enhancement as high as $6 \pm 0.2$-fold compared to the inside-out $Yb@Lu_{0.8}Er_{0.2}@Lu$ UCNPs. Given that they had essentially the same number of active ions and same interface between sensitizers and emitters, we attributed the difference in their luminescences to different distances in migration from $Yb^{3+}$ to the interface. In inside-out $Yb@Lu_{0.8}Er_{0.2}@Lu$, the energy absorbed in the core needed to travel a longer distance to the interface on average, increasing the possibility of quenching.

## Monte Carlo simulation of energy migration via $Yb^{3+}$
A back-of-the-envelope calculation, schematized in Fig. 4a, shows that the average shortest distance to the interface for sensitizers in the core is about 1.8 times longer than for those in the shell (See details in Supplementary Methods). In order to further understand the energy migration process in the above experiments, which lead to different luminescence properties for outside-in and inside-out nanoparticles with the same sensitizer-activator interface and concentrations, we carried out a series of Monte Carlo simulations of energy migration steps based on realistic $P\bar{6}$ hexagonal lattices[49–51]. Migration was only allowed between neighboring $Yb^{3+}$ sites, as shown in Fig. 4b. There exist only two types of $Yb^{3+}$ sites, namely the **1a** site situated at the vertices of the unit cell and the **1f** site inside the unit cell. Energy migration between these sites was simulated with distance-dependent probabilities based on Dexter's theory[52]. For each $Yb^{3+}$ at a **1a** site, there are two neighboring **1a** sites and three **1f** neighbors with a 12 fold higher[53] probability of going from **1a** to **1a** than from **1a** to **1f**. For each $Yb^{3+}$ at a **1f** site, there are six **1a** neighbors at equal distances, hence with equal probability of being migrated to from the **1f** site. For each $Yb^{3+}$ out of the twelve thousand ones located in the core (or in the shell, depending on the architecture), the simulation begins with a quantum of energy absorbed locally and followed by migrations amongst the sensitizers according to the above rules until its arrival at the core-shell interface. We repeated the simulation for each $Yb^{3+}$ site 1000 times, calculated the average number of migration steps from each $Yb^{3+}$ initiation site, and found that it took fewer migration steps to reach the interface when energy migration was initiated from the shell than from within the core. The migration steps histogram for all sensitizer lattice sites in the shell had a more prominent peak at low numbers of migration steps than did sites in the core (arrows in Fig. 4c). Figure 4d, e plotted the radial and elevation dependence of the migration steps for the inside-out and outside-in architectures. The migration steps were numerous at the center of the core and decreased monotonically toward the interface while the migration steps were broadly distributed radially. Migration steps were more uniformly distributed across different elevations in the core than in the shell. It took many more steps to reach the core-shell boundary when the initiation site was located at equatorial of the shell than at the poles of the shell. To better visualize the migration steps distribution, azimuthally averaged 2D heatmaps were generated and overlaid for both cases (Fig. 4f). The significant differences of energy migration via sensitizers in distinct geometrical configurations visualized by basic Monte Carlo simulation reassured the experimental evidences revealed by single-particle luminescence data.

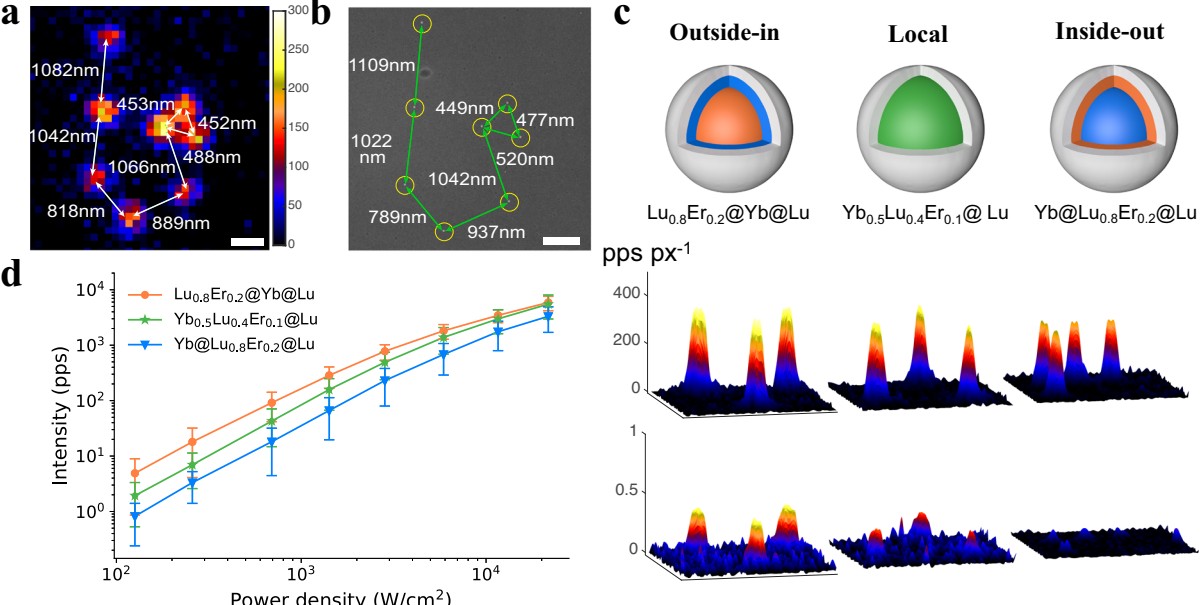

**Fig. 3 | Colocalization and single-particle imaging of outside-in, inside-out and local energy transfer core-interior shell-inert shell UCNPs. a** Wide-field luminescence image of $Lu_{0.8}Er_{0.2}@Yb@Lu$ at a power density of 21.7 kW/cm². Scare bar, 500 nm. **b** SEM image corresponding to the same region with respect to **a** Three field of views were obtained for the colocalization expriments, which showed similar results. Scale bar, 500 nm. **c** Three-dimensional representation of the wide field UCL images at 21.7 kW/cm² (top) and 126 W/cm² (bottom). **d** Saturation curves of single-particle brightness at power densities from 126 W/cm² to 21.7 kW/cm² obtained with wide-field microscopy. The results were presented as means ± standard deviation (two independent experiments, more than five field of views of wide-field images were acquired for each experiments; "n" represents the number of single nanoparticles, $Lu_{0.8}Er_{0.2}@Yb@Lu$: *n* = 116, $Yb_{0.5}Lu_{0.4}Er_{0.1}@$ Lu: n= 229, $Yb@Lu_{0.8}Er_{0.2}@Lu$: n = 45), "pps" means photons per second, and "pps px⁻¹" means photons per second per pixel. Source data are available as Source Data file.

Our simulation deals only with the short-range Dexter's exchange mechanism[52] when energy migrates along the connected lattice network[53]. Förster type multipole interactions do contribute to energy transfer when considering all possible acceptors in the vicinity of a particular donor[54–57]. The average transfer rate strongly depends on donor-acceptor distance. Considering that the exchange mechanism is dominant at short distances and other perturbations have small effect at long distances, our numerical approach is justified. More sophisticated spatial aware numerical methods combining the stochastic approach and deterministic analysis should be developed in the future to get a refined picture of how energy migrates in such a delicate system.

**Optimizing Yb³⁺ doping concentration in the interior shell**
The above demonstrated that the outside-in $Lu_{0.8}Er_{0.2}@Yb@Lu$. UCNPs had the brightest upconversion emission. To further improve the single-particle brightness, we decreased Yb³⁺ doping in the interior shell of the outside-in architecture (Fig. 5a) considering the potential concentration quenching effect[40]. Outside-in UCNPs with various concentrations of Yb³⁺ ions were synthesized. TEM images showed uniform nanoparticles, and all the UCNPs had almost the same size distribution (Supplementary Fig. 5). XRD results showed these particles maintained hexagonal phase (Supplementary Fig. 6). Ensemble measurements showed that upconversion emission decreased as the concentration of Yb³⁺ ions in the interior shell declined (Fig. 5b). The spectra exhibited a trend that higher Yb³⁺ doping concentration and/or higher power density would lead to relatively more red emission compared to green emissions. Specifically, the red emission at 654 nm dropped much more rapidly than the green emissions at 521 and 541 nm when Yb³⁺ concentration falling from 100% to 50%. In addition, the Red-to-Green ratio gradually increased with increasing excitation power density (Fig. 5c). These spectral characteristics could be

explained by the fact that the red emission process of Er³⁺ involves multiphoton process[58] and is sensitive to the power density and Yb³⁺ doping concentration[59]. The single-particle brightness data for these UCNPs is plotted in Fig. 5d. UCNPs with 100% Yb³⁺ doping in the interior shell had the most enhanced single-particle upconversion emission, around 1.7 ± 0.1 times brighter than those with 50% Yb³⁺. As the concentration of Yb³⁺ increases, there exist two competing effects, namely the improved capability of absorbing excitation energy and the heavier concentration quenching between Yb³⁺ ions. In the outside-in UCNPs, the enhanced absorption always dominated over concentration quenching at all investigated irradiances. These conclusions were also tested by Monte Carlo simulations (Fig. 5e, f). The energy migration process from each Yb³⁺ ion site to the core-interior shell interface was simulated 3000 times, and averaged migration steps for each initiation site were recorded. The migration step histogram together and cumulative frequency plot showed that it took fewer migration steps to reach the interface for almost all Yb³⁺ sites at higher Yb³⁺ doping concentrations. There were very minor exceptions for those Yb³⁺ sites sit right next to the boundary, where they preferred hopping over the interface instead of migrating away, due to reduced connectivity between Yb³⁺ sites at lower Yb³⁺ concentrations[60,61]. The simulation results were consistent with the experimental results. Therefore, pure NaYbF₄ was adopted for subsequent experiments.

**Optimizing Er³⁺ doping concentration and substrate materials**
Cross-relaxation between Er³⁺ ions may hamper UCL emission by keeping the emitter ions in an intermediate energy level[37]. In order to find the optimal Er³⁺ doping concentration for upconversion luminescence, the same three nanoparticle architectures studied above were made with 5, 10 or 30% Er³⁺ doping instead of the original 20%. TEM and XRD results demonstrated that these nanoparticles were uniform in size and were in hexagonal phase (Supplementary

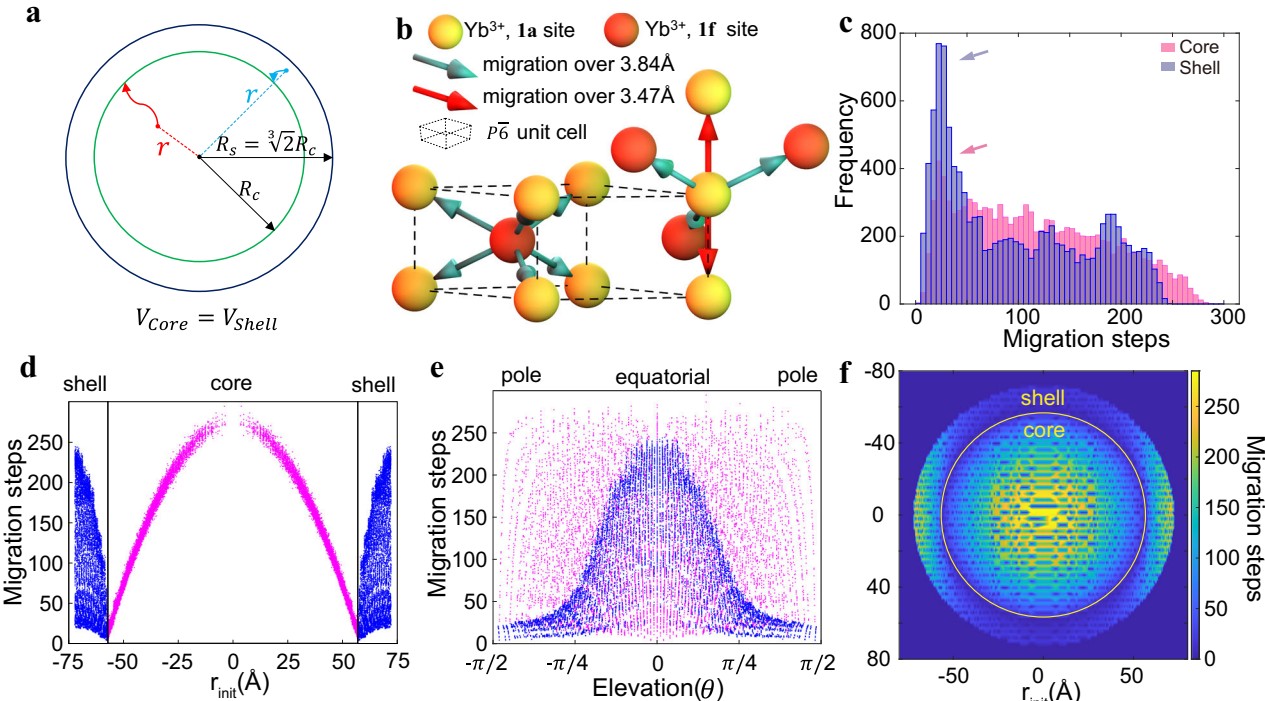

**Fig. 4 | Simulation of energy migration from Yb³⁺ to interface. a** Schematic of interior shell and corresponding radius. **b** Schematic diagram of NaYbF₄ lattice in $P\bar{6}$ hexagonal space point group. **1a** and **1f** sites are different types that possess energy migration process and the corresponding energy migration directions. **c** Frequency histograms of the number of migrations steps from core or shell to the core- interior shell interface as determined by Monte Carlo simulations with 1000 repeats. **d, e** The radial and elevation dependence of the migration steps for inside-out and outside-in topological arrangements. **f** The azimuthally averaged 2D heatmaps of migration step distribution were generated and overlaid for both inside-out and outside-in architectures.

Figs. 7–13). Single-particle imaging results showed that outside-in architectures with 5%, 10% and 30% doping had $2.8 \pm 0.1$-, $4.5 \pm 0.2$- and $4.7 \pm 0.2$-fold enhancement of luminescence under lower irradiance ($680 \, W/cm^2$) when compared with the corresponding inside-out architectures (Fig. 6a, b). Similar to the 20% Er³⁺ doping (Fig. 3d), the enhancement was also power-dependent at 10% and 30% doping. We compared the luminescence of outside-in UCNPs with different Er³⁺ doping concentrations. UCNPs with 10% Er³⁺ doping were the brightest in both ensemble (Supplementary Fig. 14) and single-particle measurements. The saturation curve of single-particle brightness is displayed in Fig. 6d. At all irradiances examined, Lu₀.₉Er₀.₁@Yb@Lu UCNPs showed the brightest single-particle emission, with a $14 \pm 0.9$-fold enhancement compared to a conventional core-shell architecture[62,63] of β-NaLu₀.₇₈Yb₀.₂Er₀.₀₂F₄@NaLuF₄ UCNPs with the same size (Fig. 6e, f, Supplementary Fig. 15).

In the above optimizations, we utilized NaLuF₄ as substrate materials. In order to investigate the generality of the topologically segregated core-shell structure strategy, we also synthesized a series of 10% Er³⁺ doping NaYF₄-based UCNPs (Y-based UCNPs), including the outside-in NaY₀.₉Er₀.₁F₄@NaYbF₄@NaYF₄ (simplified as Y₀.₉Er₀.₁@Yb@Y), inside-out NaYbF₄@NaY₀.₉Er₀.₁F₄@NaYF₄ (Yb@Y₀.₉Er₀.₁@Y), local energy transfer structure NaYb₀.₅Y₀.₄₅Er₀.₀₅F₄ @ NaYF₄ (Yb₀.₅Y₀.₄₅Er₀.₀₅@Y), and the conventional core-shell structures NaYb₀.₂Y₀.₇₈Er₀.₀₂F₄@NaYF₄ (Yb₀.₂Y₀.₇₈Er₀.₀₂@Y) (see details in Supplementary Figs. 16–20). A similar conclusion was obtained that the outside-in architecture showed the brightest upconversion emission at the single-particle level (Supplementary Fig. 20). However, the Y-based outside-in architecture Y₀.₉Er₀.₁@Yb@Y ($3572 \pm 1177$ pps at $21.7 \, kW/cm^2$) was dimmer than the Lu-based outside-in architecture Lu₀.₉Er₀.₁@Yb@Lu ($6362 \pm 2439$ pps at $21.7 \, kW/cm^2$), which may be explained by the lattice mismatch between the interior shell of NaYbF₄ and the

outmost shell of NaYF₄ in Y₀.₉Er₀.₁@Yb@Y. From the high-resolution TEM images, we found that compared to Lu₀.₉Er₀.₁@Yb@Lu, Y₀.₉Er₀.₁@Yb@Y showed a serious anisotropic growth (Supplementary Fig. 17), which could diminish the protecting effect of the outmost shell from surface quenching. Compared to Y³⁺, Lu³⁺ has cation diameter and chemical properties closer to Yb³⁺ or Er³⁺. Therefore, Lu³⁺ doping could minimize the lattice mismatch[64,65] between different layers and it was easy for us to control the nanoparticle's size and make an unbiased comparison between different structures. Therefore, Lu-based outside-in UCNPs architecture was used for the subsequently biological applications.

### Upconversion quantum yield

As a quantitative measure of upconversion efficiency, upconversion quantum yield (UCQY) characterizes the luminescence potential of upconverting materials. Only under well-defined experimental conditions and the homogeneity in size and composition of single UCNPs, one may correlate the ensemble UCQY with the single-particle brightness. We measured the absolute quantum yield for all the Lu-based UCNPs (See Supplementary Methods and Supplementary Table 4). The quantum yield results are indeed consistent with the single-particle characterization indicating the excellent homogeneity in size and composition of UCNPs used in this study. For the 10% Er³⁺ doping series with the optimal doping concentration for single-particle imaging, the outside-in structure showed the highest quantum yield of $1.9 \pm 0.4\%$, the local energy transfer structure is $0.7 \pm 0.3\%$, and the inside-out structure had the lowest quantum yield of $0.4 \pm 0.2\%$.

### Long-term single-particle tracking in living cells

The excellent photostability of UCNPs is ideal for the long-term tracking inside living cells[66]. We chose the brightest outside-in

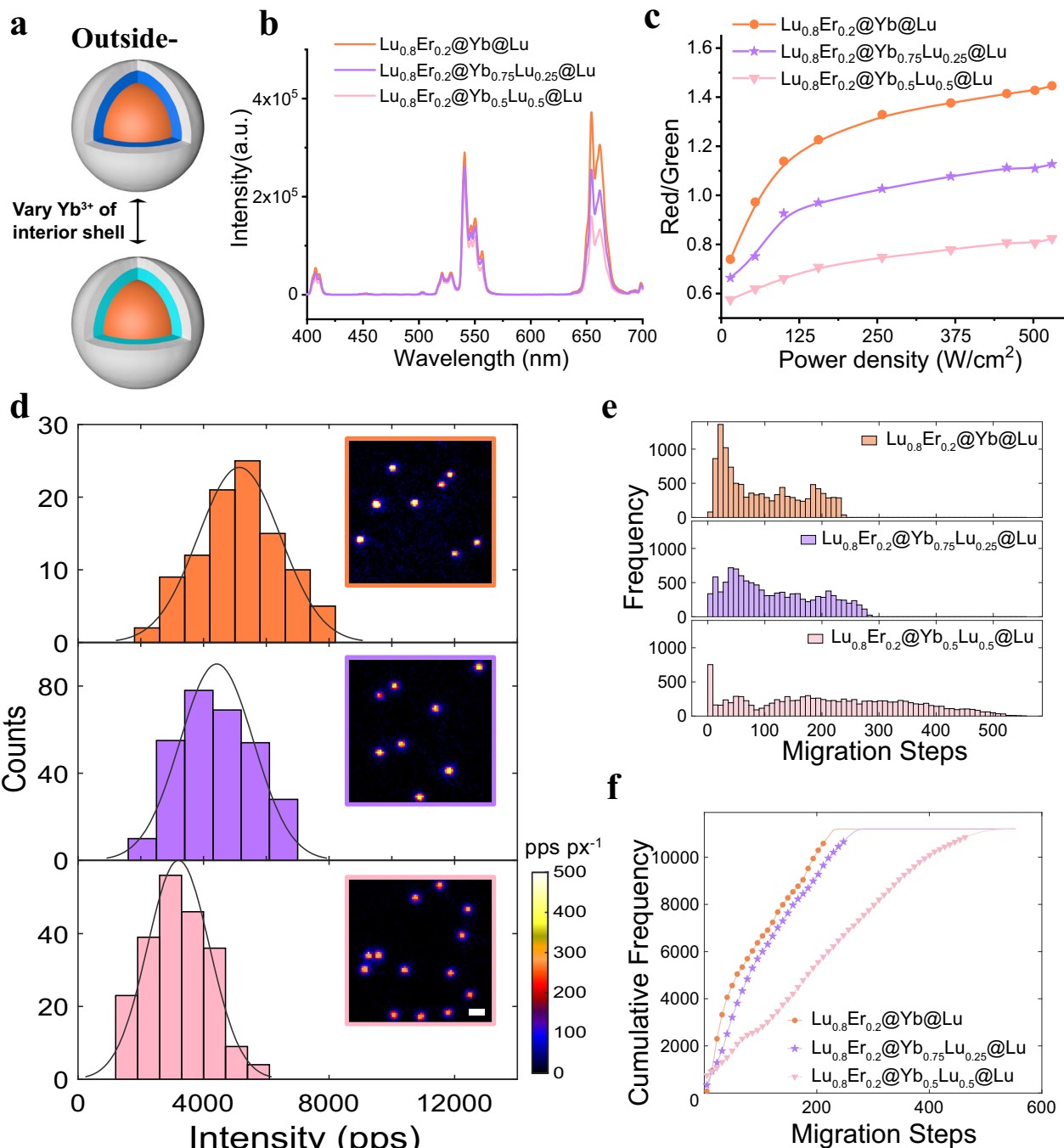

**Fig. 5 | Effect of varying the doping concentration of Yb³⁺ ions in interior shell.** **a** Schematic diagram of varying the doping concentration of Yb³⁺ ions in the interior shell. **b** UCL spectra of ensemble solution under 980 nm laser irradiation at 530 W/cm². **c** The power dependence of Red-to-Green ratio. The Red-to-Green ratio compares integrated UCL of red emission at 654 nm (from 635 to 683 nm) with that of green emission peaks at 521 nm (from 510 to 534 nm) and at 541 nm (from 535 to 577 nm). **d** Single-particle brightness distribution histogram and single-particle luminescent image of UCNPs with different Yb³⁺ ions doping at 21.7 kW/cm². Shown from top to bottom are $Lu_{0.8}Er_{0.2}$@Yb@Lu, $Lu_{0.8}Er_{0.2}$@Yb$_{0.75}$Lu$_{0.25}$@Lu and $Lu_{0.8}Er_{0.2}$@Yb$_{0.5}$Lu$_{0.5}$@Lu UCNPs, respectively. The inserts are single-particle luminescent images sharing a scale bar of 1 μm and a shared colormap was displayed at the bottom right corner. There were more than five field of views for each sample. **e** Histograms of migrations steps for all Yb³⁺ lattice sites from Monte Carlo simulations of 3000 repeats. **f** Cumulative frequency plots showing number of Yb³⁺ sites that takes less migration steps than a given number of steps for all three cases in **b**–**d**. Source data are available as Source Data file.

Lu$_{0.9}$Er$_{0.1}$@Yb@Lu UCNPs and coated with a dense silica shell (dSiO$_2$) to transfer it into the aqueous phase and improve its biocompatibility (Supplementary Fig. 21a). The conventional Yb$_{0.2}$Lu$_{0.78}$Er$_{0.02}$@Lu UCNPs with dSiO$_2$ modification was used as control (Supplementary Fig. 21b). We incubated live U2OS cancer cells with these two UCNPs, respectively, and analyzed the brightness of these probes in a cellular context. The outside-in architecture showed a significantly enhanced luminescence compared to the conventional configuration, which enabled the long-term tracking in live cells with ~15 nm localization precision and a time resolution of 10 fps (Supplementary Fig. 22a). In contrast, conventional Yb$_{0.2}$Lu$_{0.78}$Er$_{0.02}$@Lu UCNPs under the same imaging condition achieved only ~40 nm localization precision (Supplementary Fig. 22a). In order to achieve similar localization performance as the outside-in architecture, the conventional

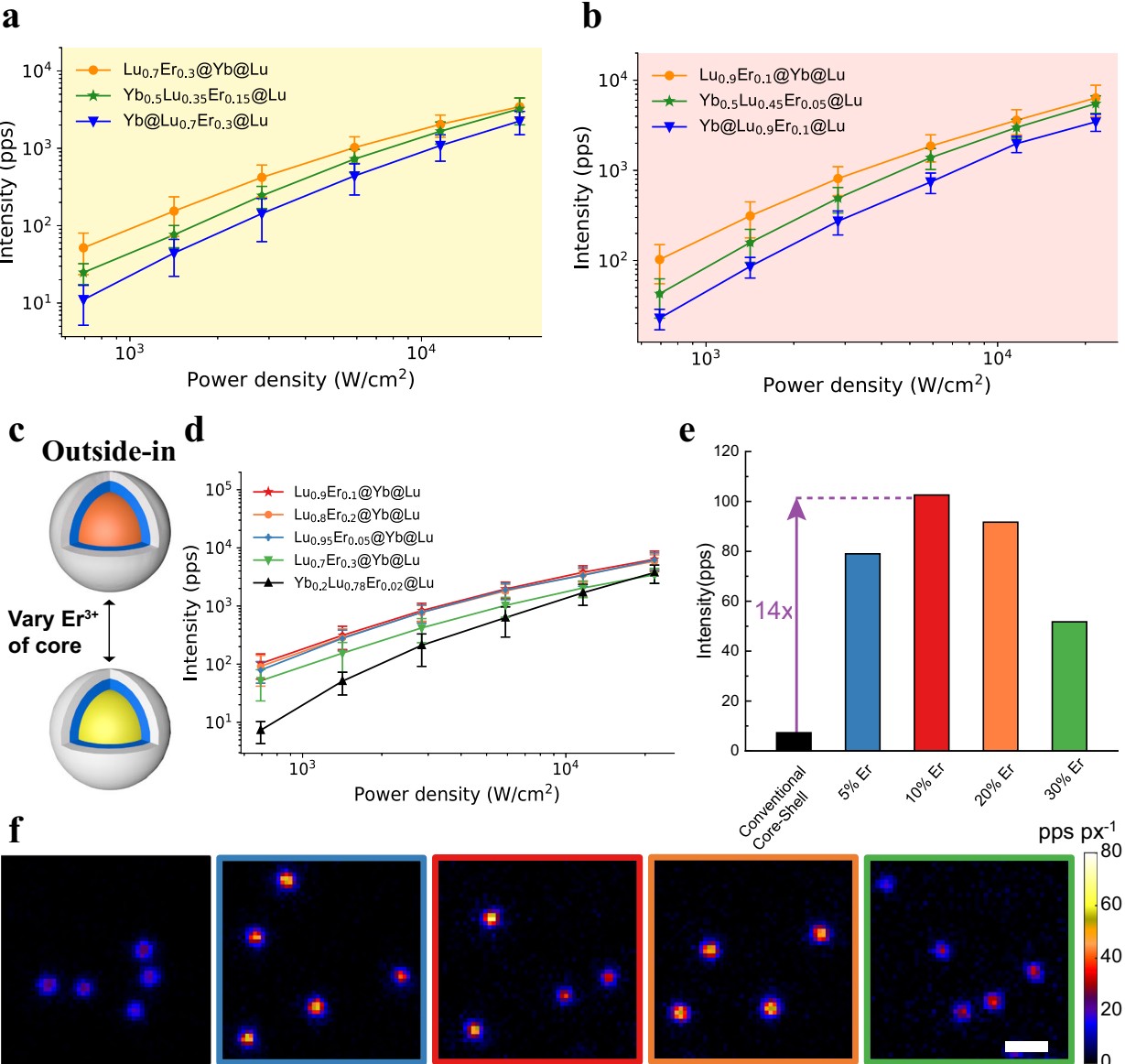

**Fig. 6 | Effect of varying the doping concentration of Er³⁺ ions in the core.**
**a** Saturation curves of single-particle brightness of UCNPs with 30% $Er^{3+}$ doping.
**b** Saturation curves of single-particle brightness of UCNPs with 10% $Er^{3+}$ doping.
**c** Schematic diagram of changing the doping concentration of $Er^{3+}$ ions in core.
**d** Saturation curves of single-particle brightness for outside-in architectures with different $Er^{3+}$ doping concentrations in core and conventional core-shell UCNPs (β-$Lu_{0.78}Yb_{0.2}Er_{0.02}$@Lu) of same size. The results of a-d were presented as means ± standard deviation (two independent experiments, more than five field of views wide-field images were acquired for each experiment; $Lu_{0.7}Er_{0.3}$@Yb@Lu: $n = 153$,

$Yb_{0.5}Lu_{0.35}Er_{0.15}$@Lu: $n = 190$, $Yb$@$Lu_{0.7}Er_{0.3}$@Lu: $n=92$, $Lu_{0.9}Er_{0.1}$@Yb@Lu: $n=219$, $Yb_{0.5}Lu_{0.45}Er_{0.05}$@ Lu: $n = 84$, $Yb$@$Lu_{0.9}Er_{0.1}$@Lu: $n = 67$, $Lu_{0.95}Er_{0.05}$@Yb@Lu: $n = 252$, $Lu_{0.78}Yb_{0.2}Er_{0.02}$@Lu: $n = 244$). **e** Single-particle brightness of conventional core-shell design and outside-in architectures UCNPs with different $Er^{3+}$ doping concentrations in core at 680 W/cm². **f** Single-particle luminescent images of conventional core-shell design and outside-in architectures UCNPs with different $Er^{3+}$ doping concentrations in core at 2.84 kW/cm². There were more than five field of views for each sample, and all single-particle luminescent images share a scale bar of 1 μm. Source data are available as Source Data file.

$Yb_{0.2}Lu_{0.78}Er_{0.02}$@Lu UCNP would have to be ~7× slower in tracking at 1.6 fps which would blur many sub-cellular activities.

As shown in Supplementary Fig. 22b–c and Supplementary Movie 1; the trajectory behaved in a "stop-and-go" fashion and appeared to be directional. This behavior may be explained by that these UCNPs were involved in active transport inside the cell[67]. We tried to fit part of the trajectory in Supplementary Fig. 22d into a hypothetical line segment and calculated the deviation of the trajectory from the line segment and found the FWHM of the deviation to be ~231 nm which was roughly on the order of the size of a cargo traveling along certain filaments such as the microtubule (Supplementary Fig. 22e). While in Supplementary Fig. 22f and Movie 2, these trajectories appeared more local and less directional representing confined

or constrained diffusion. We plotted out mean squared displacement (MSD) curves for these trajectories (Supplementary Fig. 22g) and found the average diffusion coefficients for linear segments in the MSD plot to be 0.0186 ± 0.0146 μm²/s[68]. In brief, based on the optimized outside-in UCNPs, we achieved over 10 min background free single-particle tracking with localization precision of ~15 nm, and observed two distinct modes of motion in live U2OS cells. We believed that by proper surface modification, more biological applications could be demonstrated based on our bright outside-in UCNPs.

## Discussion
In conclusion, we precisely controlled the synthesis of distinct topologically arranged architectures with outside-in, inside-out and local

energy migration and demonstrated that the outside-in architecture is the most bright in term of single-particle measurement. In order to understand the intrinsic energy migration process, a set of Monte Carlo simulations was carried out by distance-dependent random walk over $Yb^{3+}$ sites based on Dexter's theory. Simulation data reveal that the enhancement could be attributed to the fewer migration steps from sensitizer $Yb^{3+}$ to the interface in the outside-in architecture hence the minimized energy loss in migration. The single-particle brightness saturation curve showed that the enhancement was power-dependent and diminished with increasing power. We reason that there are more sensitizers in excited states at higher irradiation, reducing the effect of energy loss in migration and the quenching of back energy transfer from acceptors to sensitizers. We tuned our 17 nm outside-in architecture to optimal with 100% $Yb^{3+}$ in the interior shell and 10% $Er^{3+}$ in the core, which increased brightness 14-fold over the conventional core-shell architectures. The optimized outside-in UCNPs was further used for long-term single-particle tracking in living cells. Our investigation of energy migration in different topologically arranged architectures highlights an innovative strategy for engineering promising single-molecule imaging nanoprobes that could be translated to other applications, such as information technology, lasers manufacturing, biomedicine and bio-photonics.

## Methods

### Nanoparticle synthesis and characterization

The core-interior shell-inert shell nanoparticles were synthesized by a typical solvent thermal method, the synthesis experimental details are provided in the Supplementary Methods. The TEM images were acquired using the HT7800 software and analyzed using the ImageJ software. The ensemble measurements of upconversion properties were collected using the Fluoracle software and analyzed and visualized using the Origin software.

### Mean shortest distance to interface for sensitizers

With sensitizers and activators segregated into separate core-interior shell structures, we have two distinct topological arrangements for energy flow in the upconversion process, namely the inside-out and the outside-in configuration. Assuming that all sensitizers were uniformly distributed in their respective compartments (core or interior shell) and neglecting the finite lattice grid dimension, the shortest distance for each $Yb^{3+}$ ion to the core shell interface is the absolute difference between the radial coordinates of the sensitizers and the radius of the interface. Taking the average of the distances over all sensitizers, we can calculate the mean shortest distance for the inside-our architecture as:

$$\mathbf{d}_{\mathbf{inside\ out}}^{\mathbf{MSD}} = \frac{\int_0^{\mathbf{R_c}} \int_0^{2\pi} \int_0^{\pi} (\mathbf{R_c}-r) r^2 \sin\theta\, d\theta\, d\varphi\, dr}{\int_0^{\mathbf{R_c}} \int_0^{2\pi} \int_0^{\pi} \mathbf{r}^2 \sin\theta\, d\theta\, d\varphi\, dr} = 0.25 \mathbf{R_c} \quad (1)$$

and for the outide-in:

$$\mathbf{d}_{\mathbf{outside\ in}}^{\mathbf{MSD}} = \frac{\int_{\mathbf{R_c}}^{\mathbf{R_s}} \int_0^{2\pi} \int_0^{\pi} (r-\mathbf{R_c}) r^2 \sin\theta\, d\theta\, d\varphi\, dr}{\int_{\mathbf{R_c}}^{\mathbf{R_s}} \int_0^{2\pi} \int_0^{\pi} \mathbf{r}^2 \sin\theta\, d\theta\, d\varphi\, dr} = \left(\frac{3}{\sqrt[3]{4}} - \frac{7}{4}\right) \mathbf{R_c} \approx 0.14 \mathbf{R_c} \quad (2)$$

Hence, the mean shortest distance to interface is 1.8 times longer for sensitizers distributed in the core compared to those in the interior shell.

### Monte carlo simulations

In order to gain more insight than what we have with the naive mean shortest distance calculation, we did a series of Monte Carlo simulation to examine how energy is migrated over more realistic hexagonal lattices. We first constructed $P\bar{6}$ lattices with twelve thousand $Yb^{3+}$ sites in the core for inside-out simulation as well as roughly equal number of

$Yb^{3+}$ sites in the shell for the outside-in case. As illustrated in Fig. 3b, there were two types of lattice sites where $Yb^{3+}$ ions can occupy, namely the **1a** sites and the **1f** sites. The $Yb^{3+}$ occupancy at **1a** site is 100% while the occupancy at **1f** site is 50% with the other half occupied by sodium ions. When an energy quantum gets absorbed or arrived via migration at given $Yb^{3+}$ site, the migration probability to its neighbor is proportional to $\exp(-\frac{2r}{0.3})$ by Dexter's theory in a distance dependent fashion. Since probability of energy migrating to a **1a** site is 11.8-fold higher than migrating to a **1f** site, a random number R between 0 and 1 is employed to determine which neighbor site it migrates to in the current step as shown in Supplemental Information. Specifically, when energy is migrating out of **1a** site there will be two neighboring **1a** sites at 3.47 Å away along vertical direction and three neighboring **1f** sites 3.84 Å away on the side; when energy is migrating out of **1f** site there are six neighboring **1a** sites all at 3.84 Å away.

The simulation was repeated 1000 times for each sensitizer site as an initial energy absorption site and migrating trajectories were simulated and recorded until the migration reaches the core-shell interface. The simulation was performed using the Matlab software.

### Single-particle sample preparation

Washing the coverglass (Thermo Fisher Menzel No. 1.5 microscope coverglass, BB02400600AC13MNZ0) with 1 mL cyclohexane and dried in the air. Dropping 1 mL poly-lysine aqueous solution on the slide and holding for 1 min and then quickly washing with cyclohexane and dried in the air. And dropping 200 $\mu L$ appropriate concentration sample cyclohexane solution and holding for 1 min and then washing with cyclohexane and dried in the air.

### Single-particle imaging and data analysis

Single-particle optical characterization was performed in microscope system with Nikon 100× NA 1.49 Oil objective and a 976 nm fiber laser (BL976-PAG900, Thorlabs). And the single-particle luminescence was recorded by an EMCCD (iXon Ultra 897, Andor, Andor Solis software). The code written in Matlab is used to control laser power output. For low-power data acquisition, focus checking is important. Long exposure time and sample drifting can potentially affect the results; hence a focus check was performed by scanning a series of z-scan and reposition to the optimal focus.

Two-dimensional Gaussian fit was used to localize the particle and calculate the emission. The following formula is used for fitting,

$$I(\mathbf{x,y}) = \frac{I_0}{2\pi\sigma^2} e^{-\frac{(x-x_0)^2 + (y-y_0)^2}{2\sigma^2}} + C \quad (3)$$

where $I_0$ is the emission intensity. In order to minimize the variations from individual nanoparticles, hundreds of particles were counted and averaged according to Gaussian fitting. The analysis was performed using the Matlab software.

### Cell culture

U2OS cell line were obtained from the Cell Bank, Chinese Academy of Sciences (SCSP-5030). U2OS cells were cultured in McCoy's 5 A (modified) medium (MESGEN) supplemented with FBS (Gibco, 10%), aqueous penicillin (100 units/mL) and streptomycin (100 μg/mL), and maintained at 37 °C in a humidified atmosphere with 5% $CO_2$.

### UCNPs@dSiO₂ for long-term tracking in living cells

U2OS ($2 \times 10^4$ cells/well) was seeded in 20 mm diameter confocal dish and let grow for 24 h. UCNPs@dSiO₂ (15 nmol/mL) was added to the dish and incubated for 4 h, free UCNPs@dSiO₂ was removed by PBS buffer washing for 3 times. Afterward, the U2OS cells was imaged in the wide-field microscope system equipped with Nikon 100× NA 1.49 Oil objective and a 976 nm fiber laser (BL976-PAG900, Thorlabs).

## Reporting summary

Further information on research design is available in the Nature Research Reporting Summary linked to this article.

## Data availability

All data that support the findings of this study are presented in the manuscript and in the supplementary information file. Soure data are provided with this paper. Raw data for this study are available from the authors on request. Source data are provided with this paper.

## Code availability

The code is available from the public repository at GitHub, https://github.com/YunxiangZhangLab/LatticeHopping.

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

## Acknowledgements

This work was financially supported by the National Natural Science Foundation of China (Grant No. 22074021to Q.L. and Grant No. 22174025 to Y.Z.), Natural Science Foundation of Shanghai (Grant No. 21ZR1409100 to Y.Z.), NIH R35 grant (GM131728 to D.S.K.), Thousand Youth Talents Plan (to Q.L.), and Fudan University (start-up grant to Q.L. and Y.Z.).

## Author contributions

Q.L., Yunxiang. Z., and D.S.K. instructed this work. Yanxin. Z. carried out the nanoparticle synthesis and characterization. R.W. carried out the single-nanoparticle imaging and data analysis. Yunxiang. Z. carried out MC simulation. J.H. contributed to the photoluminescence spectrum measurement. D.G. contributed to the single-nanoparticle imaging. X.Q. contributed to the ensemble nanoparticles characterization. Q.L., D.S.K., Yunxiang. Z., Yanxin. Z., R.W. co-wrote the maunscript of the paper; all authors contributed to revising and finalizing the paper.

## Competing interests

The authors declare no competing interests.
