## [Peer Review File · Nature Communications]

Enhancement of single upconversion nanoparticle imaging by topologically segregated core-shell structure with inward energy migrationREVIEWER COMMENTS

Reviewer #1 (Remarks to the Author):

In this manuscript, the authors developed a core-shell-shell heterostructured upconversion nanoparticle (UCNP) with the composition of NaLuF₄:Er@NaYbF₄@NaLuF₄ (Outside-in) for brighter upconversion luminescence imaging of single particle, in comparison to NaYbF₄@NaLuF₄:Er@NaLuF₄, (inside -out) and NaLuF₄:Yb, Er@NaLuF₄ (local energy transfer). They discovered that the enhanced upconversion is due to the enhanced tolerance to concentration quenching in heterostructure UCNPs compared to the homogeneously doped UCNPs. In addition, compared to NaYbF₄@NaLuF₄:Er@NaLuF₄, NaLuF₄:Er@NaYbF₄@NaLuF₄ has a shorter average distance of the Yb to the interface Er ions, thus contributing to better energy transfer efficiency. They also optimized the concentration of Er and Yb in each layer and obtained an overall luminescence enhancement of 14 times compared to that of the conventional homogeneously doped core-shell UCNPs for single nanoparticle imaging. This exploration of brighter nanoparticles for single nanoparticle/molecule imaging is an interesting finding for the field. We recommend that the authors improve the manuscript by addressing the following comments. In particular, in this manuscript, the comparison was made among only these Lu doped UCNPs. As the best performing UCNPs are typically non-Lu, Y based, please compare such Lu doped UCNPs with these reported best performing conventionally used Y based similar structured UCNPs more quantitatively (e.g., the required excitation power density, the Quantum efficiencies at the ensemble level, and single-particle levels.) Therefore, the advantages of their systems can be demonstrated.

1. Line 19, in "three architectures were designed for energy transmission from Yb³⁺-Yb³⁺ to Er³⁺ within nanoparticles," the expression "Yb³⁺-Yb³⁺ to Er³⁺" is unclear.
2. Line 56, "UCNPs doped with the lanthanide ion Er³⁺ or Tm³⁺ and with Yb³⁺ as sensitizer" may lead some to misunderstand that believe that Er³⁺ or Tm³⁺ is also categorized as a sensitizer.
3. Is it possible to give a high-resolution STEM/mapping/line scan to confirm the core-shell-shell heterostructure of the UCNP?
4. What is the substrate for single-particle characterizations? The method information of single-particle characterizations does not seem to have been provided.
5. In the manuscript, the authors primarily considered the Yb transferring energy to Er at the interface, but what about the Er located inside the core? These emitters, which are much further from the sensitizer in the NaLuF₄:Er@NaYbF₄@NaLuF₄, could be less efficient for emission. Is it possible to compare the current structure with NaLuF₄@NaLuF₄:Er@NaYbF₄@NaLuF₄ constraining most of the Er at the interfacing area?
6. If possible, please include a proof-of-concept application in using these new UCNPs.
7. In this manuscript, the comparison was made among only these Lu doped UCNPs. As the best performing UCNPs are typically non-Lu, Y based, please compare such Lu doped UCNPs with these reported best performing conventionally used Y based similar structured UCNPs in a more quantitative manner (e.g., the Quantum efficiencies at the ensemble level and single-particle levels.) Therefore, the advantages of their systems can be demonstrated.
8. Figure 5 e and g : the y axis is missing. Please confirm how many particles are used for each plot.
9. Please confirm if there may be variations (compositions, brightness) among these single particles for each sample.

Reviewer #2 (Remarks to the Author):

The manuscript reports on the synthesis of three core-shell upconversion nanoparticles (NaYbF₄@NaLu_{0.8}Er_{0.2}F₄@NaLuF₄, NaLu_{0.8}Er_{0.2}F₄@NaYbF₄@NaLuF₄ and NaYb_{0.5}Lu_{0.4}Er_{0.1}F₄@NaLuF₄) demonstrating the connection between topological arrangement of the particles and energy migration, including the characterization at the single-particle level. Although I appreciate the authors' efforts and results towards the establishment of a relationship between the architecture of the nanoparticles and their upconversion efficiencies, as a whole, the manuscript reads not highly innovative, the determining element Nature Communications hunts for,

and, thus, I do not recommend its publication.

Specific comments:

1. There are numerous examples of manuscripts discussing the enhancement of the upconversion performance induced by specific designs of upconverting nanoparticles, e.g., 10.1038/s41566-021-00862-3 and 10.1038/s41467-020-14879-9 (just two examples). A paragraph summarizing these efforts and emphasizing the novelty of the proposed approach should be included in the manuscript.
2. Please define precisely what topology-dependent energy migration means.
3. Figure 2f is based on a direct comparison of luminescence intensity. Although often reported in the literature, this comparison between intensities must be done with extreme caution. Even if all the experimental conditions are the same for all the samples (and this is not mentioned in the paper) the differences in intensity might be explained by differences in the absorption coefficient of the samples. Quantitative conclusions are only extracted by measuring the emission quantum yield.
4. The explanation for the distinct slopes of the intensity-versus-power curve of Lu_{0.8}Er_{0.2}@Yb@Lu UCNP is speculative deserving more quantitative arguments.
5. Ion-Ion energy transfer simulation methods considering all the possible acceptors in the vicinity of a particular donor were also reported in other works, e.g., 10.1016/j.jlumin.2015.07.005, 10.1021/acs.jpcllett.0c03613, 10.1021/acsnanoscienceau.1c00033, just to mention a few examples. The Monte Carlo method reported here must be commented relatively to those references.
6. The part of the energy migration calculations based on Dexter-type theory together with Monte Carlo simulations describes how many hops the energy in each system (inside-out or outside-in) migrates to the interface of the core-shell structure. The authors used Dexter's theory correctly since the energy transfer between lanthanide ions at a short-range distance (< 4 Å) has the exchange mechanism (proportional to the electron densities overlap of the Yb-Yb pair) as a dominant one, not considering (correctly) long-range distances once it leads to very low energy transfer rates (see for example Table 2 for distance order from 5 (6.08 Å) to 20 (9.56 Å) in 10.1021/acs.jpcllett.0c03613). The energy transfer rates calculations are indeed not easy to evaluate, however, a discussion on the mechanisms involving Ln-Ln energy transfer will be appreciated by the readers. Thus, a brief commentary on the main physical mechanisms (i.e., dipole-dipole, dipole-quadrupole, quadrupole-quadrupole, magnetic dipole-magnetic dipole, and exchange) should be included in the manuscript (see references 10.1021/acs.jpcllett.0c03613).
7. The model is very simple with some obvious assumptions. For instance, the pathway of energy migration involving the Yb³⁺ ions located in the outside-in architecture is shorter than that of the inside-out nanoparticles because similar volumes for the core and the shell were considered (inducing, then, a small shell thickness).

Reviewer #3 (Remarks to the Author):

The manuscript by Qian Liu and coworkers addresses an important question regarding the effect of geometric arrangement on energy migration over UCNP sublattices. There were sporadic hints in other literatures that there might be some effect when tweaking contents in core-shell(s) of inorganic nanocrystals which would lead to different optical properties. However, this was the first systematic and quantitative study I have seen to date focusing on the topological effect of swapping core shell contents of equal volumes. The study was a refined one that their synthesis was able to precisely control equal split of core-shell volumes in such small nanoparticles and the major characterization was done at the single-particle level. The result of the study was also exciting that they provided solid evidences to show that the energy migration was more efficient when going inward from shell to core than an outward one going from core to shell which reminds me of an interesting analogy that detonation of atomic bombs also favors an implosive design. The clearly demonstrated topological effect has a deep philosophical root which hopefully would be appreciated by not only scientists in related fields but also the general public. I would suggest a minor revision before publication.

Here's the list of the issues that need to be addressed before publication.

1. The color scheme in Fig 1a is a bit confusing. I can see that when orangish Yb ions and blue Er ions were mixed in “local” design, the color they chose for text labeling were green which makes sense. However, in the cartoon the ions were identifiable spheres which in my opinion should not switch to different colors or colors with very strange tones. I would suggest set the ion spheres in the “local” design in same colors as the ones in “outside-in” and “inside-out” designs;
2. In Fig. 1g, the outside-in UCNPs showed more than one order of magnitude enhancement than that of inside-outs. Why is there only 6-fold enhancement in the single-particle measurements?
3. In Fig.1h and Supplementary Fig. 2d, the lifetime of inside-out and local architecture UCNPs were similar, but their intensity showed great differences. The authors need to explain this behavior.
4. In Fig 3a, the colocalization pattern looks like well matched. If I look closer at the numbers of distances, I can see they aren't really identical. Can the authors provide sound justifications in the number discrepancies?
5. In Fig 3d, units “pps”, “pps px-1” were not explained anywhere in the text or caption;
6. In Figure 6, the authors only synthesized the outside-in UCNPs with Er³⁺ doping concentration of 5%. How about the other two architectures and their structure-properties relationship?

Response to reviewer's comments

Reviewer #1 (Remarks to the Author):

In this manuscript, the authors developed a core-shell-shell heterostructured upconversion nanoparticle (UCNP) with the composition of NaLuF₄:Er@NaYbF₄@NaLuF₄ (Outside-in) for brighter upconversion luminescence imaging of single particle, in comparison to NaYbF₄@NaLuF₄:Er@NaLuF₄, (inside - out) and NaLuF₄:Yb, Er@NaLuF₄ (local energy transfer). They discovered that the enhanced upconversion is due to the enhanced tolerance to concentration quenching in heterostructure UCNPs compared to the homogeneously doped UCNPs. In addition, compared to NaYbF₄@NaLuF₄:Er@NaLuF₄, NaLuF₄:Er@NaYbF₄@NaLuF₄ has a shorter average distance of the Yb to the interface Er ions, thus contributing to better energy transfer efficiency. They also optimized the concentration of Er and Yb in each layer and obtained an overall luminescence enhancement of 14 times compared to that of the conventional homogeneously doped core-shell UCNPs for single nanoparticle imaging. This exploration of brighter nanoparticles for single nanoparticle/molecule imaging is an interesting finding in the field. We recommend that the authors improve the manuscript by addressing the following comments.

We thank the reviewer for the positive comments and helpful suggestions.

1. In particular, in this manuscript, the comparison was made among only these Lu-doped UCNPs. As the best performing UCNPs are typically non-Lu, Y based, please compare such Lu-doped UCNPs with these reported best performing conventionally used Y-based similar structured UCNPs more quantitatively (e.g., the required excitation power density, the Quantum efficiencies at the ensemble level, and single-particle levels.) Therefore, the advantages of their systems can be demonstrated.

Response: Following these suggestions, we synthesized a new series of nanoparticles with Y³⁺ replacing Lu³⁺ and investigated their optical properties at the ensemble and single particle level, respectively. We chose 10% Er³⁺ doping for the Y-based UCNPs, since we know it is the optimal doping concentration for single particle brightness. The newly synthesized nanoparticles included the outside-in structure NaY_{0.9}Er_{0.1}F₄@NaYbF₄@NaYF₄ (simplified as Y_{0.9}Er_{0.1}@Yb@Y), inside-out NaYbF₄@NaY_{0.9}Er_{0.1}F₄@NaYF₄ (Yb@Y_{0.9}Er_{0.1}@Y), local energy transfer NaYb_{0.5}Y_{0.45}Er_{0.05}F₄@NaYF₄ (Yb_{0.5}Y_{0.45}Er_{0.05}@Y), and the conventional core-shell structures NaYb_{0.2}Y_{0.78}Er_{0.02}F₄@NaYF₄ (Yb_{0.2}Y_{0.78}Er_{0.02}@Y). According to our experiments, highly doped Y³⁺, such as in NaY_{0.9}Er_{0.1}F₄ and NaYb_{0.2}Y_{0.78}Er_{0.02}F₄, induced a reduction of the core size from 11 nm to 7 nm (Figure R1) when using the same synthesizing protocol, which maybe resulted from the difference of chemical properties between Y³⁺ and Yb³⁺ or Lu³⁺. In order to make a fair comparison, we grew another layer of NaREF₄ with the same composition of the initial core (Figure R1) to make the nanoparticles' size around 11 nm.

Figure R1. TEM images of Y-based UCNPs with 10% Er³⁺ doping. TEM images of ~7 nm core of NaY_{0.9}Er_{0.1}F₄ and NaYb_{0.2}Y_{0.78}Er_{0.02}F₄ (first column), ~11 nm NaYb_{0.5}Y_{0.45}Er_{0.05}F₄, NaY_{0.9}Er_{0.1}F₄, NaYbF₄ and NaYb_{0.2}Y_{0.78}Er_{0.02}F₄ (second column), ~14 nm NaYb_{0.5}Y_{0.45}Er_{0.05}F₄, NaY_{0.9}Er_{0.1}F₄@NaYbF₄, NaYbF₄@NaY_{0.9}Er_{0.1}F₄ and NaYb_{0.2}Y_{0.78}Er_{0.02}F₄ (third column), ~17 nm NaYb_{0.5}Y_{0.45}Er_{0.05}F₄@NaYF₄, NaY_{0.9}Er_{0.1}F₄@NaYbF₄@NaYF₄, NaYbF₄@NaY_{0.9}Er_{0.1}F₄@NaYF₄ and NaYb_{0.2}Y_{0.78}Er_{0.02}F₄@NaYF₄ UCNPs (right). Each panel includes a size distribution histogram with Gaussian fitting curve; the mean size (by Gaussian fitting) and standard deviation are shown in the lower left corner of each panel. Highly doped Y³⁺, such as in NaY_{0.9}Er_{0.1}F₄ and NaYb_{0.2}Y_{0.78}Er_{0.02}F₄, induced a reduction of the core size from 11 nm to 7 nm when using the same synthesizing protocol. In order to make a fair comparison, we grew another layer of NaREF₄ with the same composition of the in initial core to make the nanoparticles' size around 11 nm.

Considering these final multi-shell Y-based UCNPs underwent different synthesis procedures, it is difficult for us to quantitate the concentration of these nanoparticles and make a fair comparison by ensemble spectra measurement. Therefore, we characterized only their lifetime at ensemble level and the brightness at single particle level. We summarized the lifetime of Y-based UCNPs in Table R1. From the lifetime and single particle brightness results (Figure R3-4), we found these three architectures showed similar results as Lu-based UCNPs, the outside-in architecture Y_{0.9}Er_{0.1}@Yb@Y showed the longest luminescent lifetime for emissions at 541 nm and 654 nm and the brightest upconversion luminescence at the single particle level, which was 2.3±0.1 times brighter than that of Yb@Y_{0.9}Er_{0.1}@Y and 2.7±0.1 times than Yb_{0.2}Y_{0.78}Er_{0.02}@Y.

Besides, the Y-based outside-in architecture Y_{0.9}Er_{0.1}@Yb@Y was dimmer than the Lu-based outside-in architecture Lu_{0.9}Er_{0.1}@Yb@Lu, which may be explained by the lattice mismatch between the interior shell of NaYbF₄ and the outmost shell of NaYF₄ in Y_{0.9}Er_{0.1}@Yb@Y. From the high-resolution TEM images, we found that compared to Lu_{0.9}Er_{0.1}@Yb@Lu, Y_{0.9}Er_{0.1}@Yb@Y showed a serious anisotropic growth (Figure R2), which could diminish the protecting effect of the outmost shell from surface

quenching. Compared to Y^{3+} , Lu^{3+} has cation diameter and chemical properties closer to Yb^{3+} or Er^{3+} . Therefore, Lu^{3+} doping could minimize the lattice mismatch (N. J. Johnson, et al. *ACS Nano* **8**, 10517-10527 (2014); Liu D, et al. *Nat. Commun.* **7**, 10254 (2016)) between different layers and it was easy for us to control the nanoparticle's size and make an unbiased comparison between different structures. Therefore, in this manuscript we focused on Lu-based UCNPs.

Figure R2. The high-resolution TEM images of $Lu_{0.9}Er_{0.1}@Yb@Lu$ (left) and $Y_{0.9}Er_{0.1}@Yb@Y$ (right). The blue outlines represent the isotropy content with a near circle shape and the orange outlines represent the anisotropy $NaYF_4$ shell due to the lattice mismatch.

Figure R3. Luminescence decay curves of Y-based UCNPs, excited by 980 nm pulsed laser and recorded at (a) 541 nm; (b) 654 nm emission.

Figure R4. Single-particle brightness of Y-based UCNPs. Saturation curves of single particle brightness at power densities from 126 W/cm² to 21.7 kW/cm² obtained with wide-field microscopy.

Table R1. Lifetime of UCNPs, which was excited under 100 W/cm² 980 nm laser and recorded at 541 nm and 654 nm emissions.

Nanoparticle	541 nm emission (μ s)	654 nm emission (μ s)
Lu _{0.95} Er _{0.05} @Yb@Lu	114.38	105.51
Yb@Lu _{0.95} Er _{0.05} @Lu	85.39	69.04
Yb _{0.5} Lu _{0.475} Er _{0.025} @Lu	52.52	72.96
Lu _{0.9} Er _{0.1} @Yb@Lu	151.41	108.93
Yb@Lu _{0.9} Er _{0.1} @Lu	72.87	66.36
Yb _{0.5} Lu _{0.45} Er _{0.05} @Lu	94.61	101.50
Lu _{0.8} Er _{0.2} @Yb _{0.75} Lu _{0.25} @Lu	125.86	148.67
Lu _{0.8} Er _{0.2} @Yb _{0.5} Lu _{0.5} @Lu	108.61	126.88
Lu _{0.7} Er _{0.3} @Yb@Lu	132.97	137.94
Yb@Lu _{0.7} Er _{0.3} @Lu	81.54	43.95
Yb _{0.5} Lu _{0.35} Er _{0.15} @Lu	81.32	64.56
Y _{0.9} Er _{0.1} @Yb@Y	201.30	178.10
Yb@Y _{0.9} Er _{0.1} @Y	45.24	61.08
Yb _{0.5} Y _{0.45} Er _{0.05} @Y	56.71	66.17

In the revised Manuscript, we added the following description in Page 12, line 341: “In the above optimizations, we utilized NaLuF₄ as substrate materials. In order to investigate the generality of the topologically segregated core-shell structure strategy, we also synthesized a series of 10% Er³⁺ doping NaYF₄-based UCNPs (Y-based UCNPs), including the outside-in NaY_{0.9}Er_{0.1}F₄@NaYbF₄@NaYF₄ (simplified as Y_{0.9}Er_{0.1}@Yb@Y), inside-out NaYbF₄@NaY_{0.9}Er_{0.1}F₄@NaYF₄ (Yb@Y_{0.9}Er_{0.1}@Y), local energy transfer structure NaYb_{0.5}Y_{0.45}Er_{0.05}F₄@NaYF₄ (Yb_{0.5}Y_{0.45}Er_{0.05}@Y), and the conventional core-shell structures NaYb_{0.2}Y_{0.78}Er_{0.02}F₄@NaYF₄

($\text{Yb}_{0.2}\text{Y}_{0.78}\text{Er}_{0.02}@Y$) (see details in Supplementary Figure 14-18). A similar conclusion was obtained that the outside-in architecture showed the brightest upconversion emission at the single particle level (Supplementary Figure 18). However, the Y-based outside-in architecture $\text{Y}_{0.9}\text{Er}_{0.1}@Yb@Y$ (3572 ± 1177 pps at 21.7 kW/cm^2) was dimmer than the Lu-based outside-in architecture $\text{Lu}_{0.9}\text{Er}_{0.1}@Yb@Lu$ (6362 ± 2439 pps at 21.7 kW/cm^2), which may be explained by the lattice mismatch between the interior shell of NaYbF_4 and the outmost shell of NaYF_4 $\text{Y}_{0.9}\text{Er}_{0.1}@Yb@Y$. From the high-resolution TEM images, we found that compared to $\text{Lu}_{0.9}\text{Er}_{0.1}@Yb@Lu$, $\text{Y}_{0.9}\text{Er}_{0.1}@Yb@Y$ showed a serious anisotropic growth (Supplementary Figure 15), which could diminish the protecting effect of the outmost shell from surface quenching. Compared to Y^{3+} , Lu^{3+} has cation diameter and chemical properties closer to Yb^{3+} or Er^{3+} . Therefore, Lu^{3+} doping could minimize the lattice mismatch^{74,75} between different layers and it was easy for us to control the nanoparticle's size and make an unbiased comparison between different structures. Therefore, Lu-based outside-in UCNPs architecture was used for the subsequently biological applications.”

We also added **Figure R1-4** to Supplementary as **Figure 14-15, 17-18; Table R1** to Supplementary as **Table 3**.

2. Line 19, in “three architectures were designed for energy transmission from Yb^{3+} - Yb^{3+} to Er^{3+} within nanoparticles,” the expression “ Yb^{3+} - Yb^{3+} to Er^{3+} ” is unclear.

Response: Thank you so much for your careful check. We tried to describe the energy transfer process initiated by Yb^{3+} ions absorbing near-infrared photons through the energy migration between Yb^{3+} ions followed by energy transfer to Er^{3+} ions until the emission of high-energy photons, just as shown in Figure 1b in the revised manuscript. In order to make a clear description, the discussion was revised to “three architectures were designed for considerations pertaining to energy migration between Yb^{3+} ions and energy transfer from Yb^{3+} to Er^{3+} within nanoparticles.” in line 19 of the revised manuscript.

3. Line 56, “UCNPs doped with the lanthanide ion Er^{3+} or Tm^{3+} and with Yb^{3+} as sensitizer” may lead some to misunderstand that believe that Er^{3+} or Tm^{3+} is also categorized as a sensitizer.

Response: The discussion was revised to “UCNPs doped with the lanthanide ion Er^{3+} or Tm^{3+} as emitter and Yb^{3+} as sensitizer.” in line 58 of the revised manuscript.

4. Is it possible to give a high-resolution STEM/mapping/line scan to confirm the core-shell-shell heterostructure of the UCNP?

Response: We greatly appreciate the valuable comments. The limitation of TEM makes it challenging to directly observe the shell formation, especially for such small UCNPs. We have tried EDX mapping for the outside-in and inside-out UCNPs (Supplementary Figure S1) in the original manuscript (Figure R5). Unfortunately, we could not tell the difference between EDX mapping results for outside-in and inside-out configurations.

We also attempted on multiple instruments which yielded similar results (see Figure R5-7). There were some literatures studying the internal element distribution through EDX and X-ray photoelectron spectroscopy (XPS), however both characterization methods have size limitations. For nanoparticles size greater than 30 nm, they can get a distinct result of the core-shell distribution of different elements through EDX and line scan, which can be also used to prove the uniform distribution of co-doped sub-30 nm UCNPs (Prigozhin MB, et al. *Nat. Nanotechnol* **14**, 420-425 (2019); Nampi PP, et al. *Mater. Sci. Eng. C Mater. Biol. Appl.* **124**, 111937 (2021); Hudry D, et al. *Small* **17**, e2104441 (2021); Mendez-Gonzalez D, et al. *Small* **18**, 2105652 (2021). And the XPS was applied to nearly 50 nm UCNPs to confirm the core-shell heterostructure (Abel K, et al. *J. Am. Chem. Soc* **131**, 5533 (2010); Xu X, et al. *Nanoscale* **9**, 7719-7726 (2017); Clark PCJ, et al. *Small*, e2107976 (2022)). However, the size of our particles is too small (~17 nm) and such small nanoparticles would easily be broken down at high voltage.

Figure R5. SEM-EDS mapping images of outside-in and inside-out UCNPs with 20% Er^{3+} doping. These images were carried out in a FEI Tecnai G2 F20 X-TWIN TEM (at Fudan University).

Figure R6. (a) High-resolution TEM images, (b) High angle annular dark field (HAADF) scanning TEM (STEM) imaging of $\text{Lu}_{0.9}\text{Er}_{0.1}@Yb@Lu$. (c) Line scan of a single particle of $\text{Lu}_{0.9}\text{Er}_{0.1}@Yb@Lu$. (d-f) EDX mapping imaging corresponding to the same region with respect to (a-b). These images were carried out in a JEM-F200(URP) TEM (at Shiyanjia Lab (www.shiyanjia.com)).

Figure R7. HAADF-STEM image and EDX elemental mapping of $\text{Lu}_{0.9}\text{Er}_{0.1}@Yb@Lu$ was carried out in another FEI Tecnai G2 F20 X-TWIN TEM. (at Nanjing University of Posts and Telecommunications).

Figure R8. HAADF-STEM image and EDX elemental mapping of $Y_{0.9}Er_{0.1}@Yb@Y$. (a) High-resolution TEM images, (b) High angle annular dark field (HAADF) scanning TEM (STEM) imaging of $Y_{0.9}Er_{0.1}@Yb@Y$. (c) Line scan of a single particle of $Y_{0.9}Er_{0.1}@Yb@Y$. (d-f) EDX mapping imaging corresponding to the same region with respect to (a-b). These above images were carried out in a JEM-F200(URP) TEM (at Shiyanjia Lab, (www.shiyanjia.com)).

We also tried to carry out EDX characterization on the newly synthesized Y-based particles $Y_{0.9}Er_{0.1}@Yb@Y$ and the results are shown in Figure R8. From the high-resolution TEM and HAADF results (Figure R8 a-b), the $NaYF_4$ inert shell can be directly observed through the contrast due to the big difference of atomic number between Y^{3+} and Yb^{3+} (Yamashita S, et al. *Sci Rep* **8**, 12325 (2018)), which can be a piece of direct evidence to confirm the core-shell-shell heterostructure of the UCNPs. However, we also tried EDX matches of this UCNPs. The elemental distribution was random, and difficult to distinguish layers through their unique rare earth element (Figure R8 d-g). We also take the line scan of several single particles, but the elemental distributions still do not display a clear hierarchical distribution (Figure R8 c).

The limitation of available characterization technique makes it difficult to directly observe the shell formation, especially for smaller UCNPs. However, combining optical properties and particles' size changing has been widely used to characterize the formation of shell (P. Pei, et al. *Nat. Nanotechnol.* **16**, 1011-1018 (2021); C. Siefe, et al. *J. Am. Chem. Soc.* **141**, 16997-17005 (2019); S. Han, et al. *Nat. Commun.* **12**, 3704-3712 (2021)).

We added **Figure R8** to Supplementary as **Figure 16**.

5. What is the substrate for single-particle characterizations? The method information of single-particle characterizations does not seem to have been provided.

Response: Thank you so much for your careful check. The substrate for single-particle

characterizations was coverglass with a thickness of 170 μm (Thermo Fisher Menzel No. 1.5 microscope coverglass, BB02400600AC13MNZ0). The detail was added in the line 489 of the manuscript “Washing the coverglass (Thermo Fisher Menzel No. 1.5 microscope coverglass, BB02400600AC13MNZ0) with 1 mL cyclohexane and dried in the air.”

6. In the manuscript, the authors primarily considered the Yb transferring energy to Er at the interface, but what about the Er located inside the core? These emitters, which are much further from the sensitizer in the $\text{NaLuF}_4:\text{Er}@\text{NaYbF}_4@\text{NaLuF}_4$, could be less efficient for emission. Is it possible to compare the current structure with $\text{NaLuF}_4@\text{NaLuF}_4:\text{Er}@\text{NaYbF}_4@\text{NaLuF}_4$ constraining most of the Er at the interfacing area?

Response: We agree with the reviewer’s comments. An inert core was introduced in the outside-in structure to constrain the Er at the interfacing area, two core-shell-shell-shell UCNPs $\text{NaYF}_4@\text{NaLu}_{0.9}\text{Er}_{0.1}\text{F}_4@\text{NaYbF}_4@\text{NaLuF}_4$ (simplified as $\text{Y}@\text{Lu}_{0.9}\text{Er}_{0.1}@\text{Yb}@\text{Lu}$) and $\text{NaYF}_4@\text{NaLu}_{0.85}\text{Er}_{0.15}\text{F}_4@\text{NaYbF}_4@\text{NaLuF}_4$ (simplified as $\text{Y}@\text{Lu}_{0.85}\text{Er}_{0.15}@\text{Yb}@\text{Lu}$) were synthesized. It is easy for us to obtain small-size and well-crystalized NaYF_4 nanoparticles. Therefore, we adopted 7 nm NaYF_4 as the inert core, which has been demonstrated as a good template for epitaxial growth by our previous work (Liu Q, et al. *Nat. Photonics* **12**, 548-553 (2018)). The total core-shell-shell-shell nanoparticle’s size was around 17.0 nm (Figure R9). In order to make a systematic comparison, we tuned the doping concentration of Er^{3+} in the first layer of the shell. $\text{Y}@\text{Lu}_{0.9}\text{Er}_{0.1}@\text{Yb}@\text{Lu}$ has the same concentration of Er^{3+} with $\text{Lu}_{0.9}\text{Er}_{0.1}@\text{Yb}@\text{Lu}$ at the interfacing area. $\text{Y}@\text{Lu}_{0.85}\text{Er}_{0.15}@\text{Yb}@\text{Lu}$ has nearly identical number of Er^{3+} ions with that of $\text{NaLu}_{0.9}\text{Er}_{0.1}\text{F}_4@\text{NaYbF}_4@\text{NaLuF}_4$ in individual nanoparticle.

Figure R9. TEM images of Y-core UCNPs. TEM images of ~ 7 nm NaYF_4 core (left), ~ 11 nm core-interior shell (second column), ~ 14.5 nm core-interior shell-outer shell (third column), and final core-interior shell-outer shell-inert shell UCNPs (right). Each panel includes a size distribution histogram with Gaussian fitting curve; the mean size (by Gaussian fitting) and standard deviation are shown in the lower left corner of each panel.

Figure R10. Single-particle brightness of Y-core UCNPs and Lu_{0.9}Er_{0.1}@Yb@Lu. Saturation curves of single particle brightness at power densities from 126 W/cm² to 21.7 kW/cm² obtained with wide-field microscopy.

At single particle level, the Lu_{0.9}Er_{0.1}@Yb@Lu showed the brightest upconversion emission when compared to Y@Lu_{0.9}Er_{0.1}@Yb@Lu and Y@Lu_{0.85}Er_{0.15}@Yb@Lu. However, under lower irradiance (680 W/cm²), Y@Lu_{0.9}Er_{0.1}@Yb@Lu showed a similar single particle brightness with that of Lu_{0.9}Er_{0.1}@Yb@Lu. As the power density increases, the advantage of Lu_{0.9}Er_{0.1}@Yb@Lu became more significant. Therefore, we think constraining the active ions at the interface area is good for the low irradiance applications. At the high-power density, the Er³⁺ ions located at the center of the core could also be beneficial for upconversion emission. As for Y@Lu_{0.85}Er_{0.15}@Yb@Lu, whose Er³⁺ ions maintained nearly the same number as those from of Lu_{0.9}Er_{0.1}@Yb@Lu and were constrained closer to the interfacing area, the dramatic decline of brightness may be due to the cross-relaxation between Er³⁺ ions. These results confirm the efficiency of interface energy transfer and demonstrated the advantage of our architecture design.

7. *If possible, please include a proof-of-concept application for using these new UCNPs.*

Response: We gratefully appreciate your valuable suggestion. We modified the brightest outside-in UCNPs (NaLu_{0.9}Er_{0.1}F₄@NaYbF₄@NaLuF₄) and the conventional architecture (NaYb_{0.2}Lu_{0.78}Er_{0.02}F₄@ NaLuF₄) with SiO₂ to increase their water solubility and study the long-term tracking in live U-2 OS cancer cells. The experimental detail and results were added in the revised Manuscript and Supplementary, as follows.

In the revised Supplementary, Page 2:

“Synthesis of UCNPs@dSiO₂”

UCNP@dSiO₂ was synthesized via water-in-oil reverse microemulsion². 1 g Igepal CO-520 was dispersed in 10 mL of cyclohexane and sonicated for 5 min. 0.5 mL

UCNPs cyclohexane solution was then added into the mixture and stirred for 3 hours. Subsequently, 50 μL of ammonia (30%) was added dropwise and stirred for 24 hours. Last, 5 μL of tetraethyl orthosilicate (TEOS) sonicated 30 min with 200 μL cyclohexane was slowly introduced into the reaction system. After keeping magnetic stirring for 24 hours, the resulting products were precipitated by adding acetone and then washed three times with ethanol and dispersed in 10 ml of deionized water ($\text{DI H}_2\text{O}$).

Incubation of live cells with UCNPs@dSiO₂ for long-term tracking³

U-2 OS (2×10^4 cells/well) was seeded in 20 mm diameter confocal dish and let grow for 12 h. UCNPs@dSiO₂ (15 nmol/mL) was added to the dish and incubated for 4 h, UCNPs@dSiO₂ was removed and washed by PBS buffer 3 times. Afterward, the U-2 OS cells was imaged in the wide-field microscope system equipped with Nikon 100 \times NA 1.49 Oil objective and a 976 nm fiber laser (BL976-PAG900, Thorlabs).”

Figures R11-12 were added as **Supplementary Figures 19-20** in the revised Supplementary, Page 25-27:

Figure R11. TEM images of (a) $\text{Lu}_{0.9}\text{Er}_{0.1}\text{@Yb@Lu@dSiO}_2$ and (b) the conventional $\text{Yb}_{0.2}\text{Lu}_{0.78}\text{Er}_{0.02}\text{@Lu@dSiO}_2$.

Figure R12. (a) Precision frequency density distribution of $\text{Lu}_{0.9}\text{Er}_{0.1}@Yb@Lu@dSiO_2$ and $Yb_{0.2}\text{Lu}_{0.78}\text{Er}_{0.02}@Lu@dSiO_2$. (b) Bright field image of a U-2 OS cells loaded with $\text{Lu}_{0.9}\text{Er}_{0.1}@Yb@Lu@dSiO_2$. (c) Time-lapsed single particle images inside blue rectangle in (b) for a period of 60 secs with two particle trajectories overlaid, the direction is marked by the green arrows. (d) The part of trajectory distribution (yellow box in (c)) and the fitted hypothetical line segment. (e) Deviation of the trajectory from the line segment. (f) $\text{Lu}_{0.9}\text{Er}_{0.1}@Yb@Lu@dSiO_2$ loaded U-2 OS cell with trajectories of particles moving randomly in different directions. (g) Representative MSD curves plot for these trajectories of (f).

We added the following content in line 388 of the revised Manuscript:

“Long-term single-particle tracking in living cells. The excellent photostability of UCNPs is ideal for the long-term tracking inside living cells⁷⁶. We chose the brightest outside-in Lu_{0.9}Er_{0.1}@Yb@Lu UCNPs and coated with a dense silica shell (dSiO₂) to transfer it into the aqueous phase and improve its biocompatibility (Supplementary Figure 19a). The conventional Yb_{0.2}Lu_{0.78}Er_{0.02}@Lu UCNPs with dSiO₂ modification was used as control (Supplementary Figure 19b). We incubated live U-2 OS cancer cells with these two UCNPs, respectively, and analyzed the brightness of these probes in a cellular context. The outside-in architecture showed a significantly enhanced luminescence compared to the conventional configuration, which enabled the long-term tracking in live cells with ~15 nm localization precision and a time resolution of 10 fps (Supplementary Figure 20a). In contrast, conventional Yb_{0.2}Lu_{0.78}Er_{0.02}@Lu UCNPs under the same imaging condition achieved only ~40 nm localization precision (Supplementary Figure 21a). In order to achieve similar localization performance as the outside-in architecture, the conventional Yb_{0.2}Lu_{0.78}Er_{0.02}@Lu UCNP would have to be ~7x slower in tracking at 1.6 fps which would blur many sub-cellular activities.

As shown in Figure Supplementary Figure 20 (b-c) and Supplementary Movie 1; the trajectory behaved in a “stop-and-go” fashion and appeared to be directional. This behavior may be explained by that these UCNPs were involved in active transport inside the cell⁷⁷. We tried to fit part of the trajectory in Supplementary Figure 20 d into a hypothetical line segment and calculated the deviation of the trajectory from the line segment and found the FWHM of the deviation to be ~231nm which was roughly on the order of the size of a cargo traveling along certain filaments such as the microtubule (Supplementary Figure 20 e). While in Supplementary Figure 20 f and Movie 2, these trajectories appeared more local and less directional representing confined or constrained diffusion. We plotted out MSD curves for these trajectories (Supplementary Figure 20 g) and found the average diffusion coefficients for linear segments in the MSD plot to be $0.0186 \pm 0.0146 \mu\text{m}^2/\text{s}$ ⁷⁸. In brief, based on the optimized outside-in UCNPs, we achieved over 10 min background free single particle tracking with localization precision of ~15 nm, and observed two distinct modes of motion in live U-2 OS cells. We believed that by proper surface modification, more biological applications could be demonstrated by our bright outside-in UCNPs.”

8. In this manuscript, the comparison was made among only these Lu doped UCNPs. As the best performing UCNPs are typically non-Lu, Y based, please compare such Lu doped UCNPs with these reported best performing conventionally used Y based similar structured UCNPs in a more quantitative manner (e.g., the Quantum efficacies at the ensemble level and single-particle levels.) Therefore, the advantages of their systems can be demonstrated.

Response: We synthesized a new series of nanoparticles with Y³⁺ replacing Lu³⁺ and investigated their optical properties at the ensemble and single particle level,

respectively. The details were shown in the Response to Question 1.

9. *Figure 5 e and g: the y axis is missing. Please confirm how many particles are used for each plot.*

Response: The tick labels of the y axis were added in Figure 5 d and e in the revised Manuscript.

The y axis of single particle brightness histogram represents the number of particles at corresponding intensity data. We analyzed more than 100 nanoparticles for each sample to build the single particle brightness histogram in Figure 5d.

In our simulation, the nanoparticle size was fixed and only ions occupying each of the $n_l \sim 11k$ lattice sites in the interior shell could be different. The lattice configuration was always the same for an interior shell with 100% Yb^{3+} ; while 75% or 50 % of the lattice sites in the interior shell were randomly occupied with Yb^{3+} ions for the other two cases where Lu^{3+} ions were doped in the interior shells. Therefore, we simulated 3000 identical nanoparticles for the 100% Yb^{3+} case, 4000 75% Yb^{3+} nanoparticles with each site having 75% probability of being occupied with Yb^{3+} and 6000 50% Yb^{3+} nanoparticles with each site having 50% probability of being occupied with Yb^{3+} such that 3000 migration trajectories per each lattice site in the interior shell were generated for all three cases. Note each photon led to one simulated migration trajectory associated with the lattice site where it was absorbed. Hence a fair comparison can be carried out with roughly 33 million ($\sim 3000 \times 11k$) migration trajectories each for each of the three UCNPs with different Yb^{3+} doping.

10. *Please confirm if there may be variations (compositions, brightness) among these single particles for each sample.*

Response: We greatly appreciate the valuable comments. Even though we tried to control the synthesis temperature and time as precisely as possible to form unity nanoparticles, there must exist a Gaussian type of distribution in the particle size and the error of the diameter is around 1 nm. Each layer of nanoparticles also has small variations among them. The single-particle measurements reveal the individuality of each nanoparticle and give the best possible characterization for each of them. Meanwhile, by characterizing a great number of nanoparticles, we could utilize the statistical analysis to extract the averaged luminescence properties for each sample. These averaged brightness per nanoparticle better represents the luminescence nature for each sample than what ensemble measurement could offer.

To clarify this, we revised the discussion from “Hundreds of particles were fit to perform statistical analysis.” to “In order to minimize the variations from individual nanoparticles, hundreds of particles were counted and averaged according to Gaussian fitting.” in the line of 508.

Reviewer #2 (Remarks to the Author):

The manuscript reports on the synthesis of three core-shell upconversion nanoparticles (NaYbF₄@NaLu_{0.8}Er_{0.2}F₄@NaLuF₄, NaLu_{0.8}Er_{0.2}F₄@NaYbF₄@NaLuF₄, and NaYb_{0.5}Lu_{0.4}Er_{0.1}F₄@NaLuF₄) demonstrating the connection between topological arrangement of the particles and energy migration, including the characterization at the single-particle level.

Although I appreciate the authors' efforts and results towards the establishment of a relationship between the architecture of the nanoparticles and their upconversion efficiencies, as a whole, the manuscript reads not highly innovative, the determining element Nature Communications hunts for, and, thus, I do not recommend its publication.

We thank the reviewer for the helpful comments. Our manuscript focused on the relationship among the architecture of the nanoparticles, energy migration, and upconversion brightness at the single particle level. Due to the excellent photostability and background-free features, single molecule imaging based on lanthanide-doped upconversion nanoparticles has attracted great attention, which make it possible to obtain super long-term single-molecule tracking. However, they were restricted by the luminescence brightness of a single nanoparticle, especially when their size is less than 20 nm. Here, we systematically investigated the relationship between the core-shell topological arrangements and upconversion luminescence properties on 17 nm UCNPs at the single particle level. Only by swapping the core and the shell from inside-out energy migration to outside-in, the single particle brightness of UCNPs could be enhanced around 6-fold. We believe our research provided a state-of-the-art quantitative insight toward the design and synthesis of small UCNPs for bioimaging applications.

Specific comments:

1. There are numerous examples of manuscripts discussing the enhancement of the upconversion performance induced by specific designs of upconverting nanoparticles, e.g., 10.1038/s41566-021-00862-3 and 10.1038/s41467-020-14879-9 (just two examples). A paragraph summarizing these efforts and emphasizing the novelty of the proposed approach should be included in the manuscript.

Response: Thanks for the reviewer's comment and suggestion. As the reviewer mentioned, there have been many literatures, which developed methods to improve the upconversion luminescence, such as adding an outmost inert shell, optimizing the doping concentration of active ions, and introducing organic molecules to enhance light harvest ability or energy conversion efficiency (Nadort A, et al. *Nanoscale* **8**, 13099-13130 (2016); Hudry D, et al. *Adv Mater* **31**, e1900623 (2019); Zhu X, et al. *Adv Sci (Weinh)* **6**, 1901358 (2019); Chen B, et al. *Acc. Chem. Res.* **53**, 358-367 (2020); Zhou B, et al. *Nat Commun* **11**, 1174 (2020); Xu H, et al. *Nature Photonics* **15**, 732-737 (2021)). Most of them are based on the ensemble measurement. These developed strategies were inspiring but cannot be translated to single-particle's brightness optimization directly. And in most cases, these two are contradictory. For example, at

single particle level, the high doping of Er^{3+} was brighter than the ones with lower Er^{3+} doping (Gargas DJ, et al. *Nat. Nanotechnol.* **9**, 300-305 (2014)), which contrasts with observations in ensemble measurements. In addition, we found the UCNPs with inert shell performed outstanding at ensemble measurement while UCNPs with sensitizer Yb^{3+} doped outmost shell emitted more photons at single particle level (Hu J, et al. *Adv. Opt. Mater.* **10**, 2101763 (2022)). These could be explained by the difference of the excitation power densities used. For ensemble measurements and applications, relatively low power densities were used, usually from mW cm^{-2} to W cm^{-2} . For single particle imaging, a higher power density ranging from W cm^{-2} to MW cm^{-2} was adopted. Therefore, investigating the upconversion properties at single particle level is critical and necessary for single molecule imaging.

We summarized the efforts to optimize upconversion luminescence brightness and add it to the introduction in the revised Manuscript, as follows:

“Various strategies, such as adding an outmost inert shell, optimizing the doping concentration of active ions, and introducing organic molecules to enhance light harvest ability or energy converted efficiency, have been developed to optimize the upconversion³⁵⁻⁴⁰. However, most of them are based on the ensemble measurement. These developed strategies based on ensemble measurement were inspiring but cannot be translated to single-particle’s brightness optimization directly. And in most cases, these two are contradictory⁴¹.”

2. Please define precisely what topology-dependent energy migration means.

Response: Thanks for the reviewer’s suggestion. The core and shell represent simple but distinct topologies mathematically. To study the effect about the topology difference, we designed the UCNPs to have the exact same volumes for core and shell such that the same number of sensitizers or emitters can be set to occupy topological form while maintaining the exact same doping level. The energy migration is heavily influenced by the topology adopted by the sensitizers. Here we study the energy migration process through the microcosmic view, assign the Yb^{3+} ions in $P\bar{6}$ hexagonal lattices as migration sites and restrict the energy migration step by step. In our design, Yb^{3+} ions as sensitizers have very different spatial distributions, and the migration direction prefers in the less distant axial direction, which is shown in figure 4b **1a** to **1a** sites. By setting every site of Yb^{3+} ions as an initiation site, and then carry out the conditional Monte Carlo simulation along with the lattice network until reaches the interface. These architectures have a different distribution of active ions and the migration direction, as well as the steps it takes to reach interface in the macro view, are diverse.

Therefore, we define the topology-dependent energy migration as “The energy migration pattern manifested in the hopping of energy quanta amongst lattice sites occupied by lanthanide ions depends on the topology for the lattice network of sensitizers ions as well as the spatial distribution of active ions, therefore, deserves close investigation.” and the content was added in line 46 of the revised manuscript.

3. Figure 2f is based on a direct comparison of luminescence intensity. Although often reported in the literature, this comparison between intensities must be done with

extreme caution. Even if all the experimental conditions are the same for all the samples (and this is not mentioned in the paper) the differences in intensity might be explained by differences in the absorption coefficient of the samples. Quantitative conclusions are only extracted by measuring the emission quantum yield.

Response: Thanks for the reviewer's valuable suggestion.

We added the sentence "The measurement conditions for ensemble characterizations are the same for all the samples such that the results can be compared across various samples." in the revised Manuscript at line 151.

In our design, the core size was around 11 nm (volume was $\sim 700 \text{ nm}^3$) and after coating the interior shell, the nanoparticles' size was increased to around 14 nm, and the interior shell volume was also around 700 nm^3 . Based on the identical volume of the core and the interior shell, we can keep that the UCNPs have the same active volume when we swap the core and the interior shell from inside-out to outside-in structure or make the mixture structure of the core and the interior shell. In addition, ICP-MS measurement also demonstrated that the UCNPs with outside-in, inside-out, and local energy transfer structures have an identical doping concentration of sensitizer Yb^{3+} , which can absorb 980 nm photons and transfer to an emitter of Er^{3+} . Therefore, we think the sensitizer's number of each nanoparticle should be identical, the differences in the single-particle luminescence intensity didn't result from the differences in the absorption coefficient of the samples.

According to the reviewer's suggestion, we measured the absolute quantum yield for all the Lu-based UCNPs in our revised Manuscript (Table R2). The quantum yield results are consistent with that of single-particle characterization. For the 10% Er^{3+} doping series, the optimal doping concentration for single particle imaging, the outside-in structure showed the highest quantum yield of $1.87 \pm 0.35\%$, the local energy transfer structure is $0.67 \pm 0.25\%$, and the inside-out structure had the lowest quantum yield of $0.38 \pm 0.19\%$.

We added the experimental detail in the revised Supplementary Note 3 (Page 7), as follows:

“Upconversion quantum yield (UCQY) measurement:

The methods were adapted from the design of Veggel et al⁶. We perform the measurements from Edinburgh Instruments LFSP920 luminescence spectrometer modified with NIR PMT (HAMAMATSU, C9940-02, No. CA0142) and tested with an integrating sphere. The 120 W/cm^2 excitation power density was chosen and un-doped NaYF_4 UCNPs were chosen as the reference sample to obtain absolute upconversion quantum yield. The UCQY value was calculated through the following equation⁷ where the absolute UCQY is determined by first acquiring visible photons emitted (N_{em}), which is done by measuring the integrated emission intensity ($I_{\text{Sample}, 400 \text{ nm to } 700 \text{ nm}}$), and dividing its value by the 980 nm photons absorbed (N_{abs}). The latter is subtracting the intensity of the excitation beam after propagating through the sample under review (T_{Sample}) from the intensity of the beam after passing through an equivalent undoped reference sample ($T_{\text{Reference}}$), possessing similar scattering properties.”

$$\text{UCQY} = \frac{\text{visible photons emitted}}{980 \text{ nm photons absorbed}} = \frac{N_{\text{em}}(\lambda_{\text{ex}})}{N_{\text{abs}}(\lambda_{\text{ex}})} = \frac{I_{\text{Sample}}}{T_{\text{Reference}} - T_{\text{Sample}}}$$

Table R2. The summary of UCQY for Lu-based samples.

Nanoparticle	Type	UCQY
Lu _{0.95} Er _{0.05} @Yb@Lu	Outside-in	1.59±0.41%
Yb@Lu _{0.95} Er _{0.05} @Lu	Inside-out	0.42±0.23%
Yb _{0.5} Lu _{0.475} Er _{0.025} @Lu	Local energy transfer	0.84±0.36%
Lu _{0.9} Er _{0.1} @Yb@Lu	Outside-in	1.87±0.35%
Yb@Lu _{0.9} Er _{0.1} @Lu	Inside-out	0.38±0.19%
Yb _{0.5} Lu _{0.45} Er _{0.05} @Lu	Local energy transfer	0.67±0.25%
Lu _{0.8} Er _{0.2} @Yb@Lu	Outside-in	1.64±0.45%
Yb@Lu _{0.8} Er _{0.2} @Lu	Inside-out	0.35±0.18%
Yb _{0.5} Lu _{0.4} Er _{0.1} @Lu	Local energy transfer	0.59±0.21%
Lu _{0.8} Er _{0.2} @Yb _{0.75} Lu _{0.25} @Lu	Outside-in, interior shell	1.47±0.33%
Lu _{0.8} Er _{0.2} @Yb _{0.5} Lu _{0.5} @Lu	Outside-in, interior shell	1.28±0.27%
Lu _{0.7} Er _{0.3} @Yb@Lu	Outside-in	1.21±0.35%
Yb@Lu _{0.7} Er _{0.3} @Lu	Inside-out	0.28±0.11%
Yb _{0.5} Lu _{0.35} Er _{0.15} @Lu	Local energy transfer	0.51±0.22%
Yb _{0.2} Lu _{0.78} Er _{0.02} @Lu	traditional	0.73±0.35%

In the revised Manuscript, line 375 we added the description about quantum yield, as follows:

“Upconversion quantum yield. As a quantitative measure of upconversion efficiency, upconversion quantum yield (UCQY) characterizes the luminescence potential of upconverting materials. Only under well-defined experimental conditions and the homogeneity in size and composition of single UCNPs, one may correlate the ensemble UCQY with the single particle brightness. We measured the absolute quantum yield for all the Lu-based UCNPs (See Supplementary Note 3 and Supplementary Table 4). The quantum yield results are indeed consistent with the single-particle characterization indicating the excellent homogeneity in size and composition of UCNPs used in this study. For the 10% Er³⁺ doping series with the optimal doping concentration for single particle imaging, the outside-in structure showed the highest quantum yield of 1.87±0.35%, the local energy transfer structure is 0.67±0.25%, and the inside-out structure had the lowest quantum yield of 0.38±0.19%.”

4. The explanation for the distinct slopes of the intensity-versus-power curve of Lu_{0.8}Er_{0.2}@Yb@Lu UCNP is speculative deserving more quantitative arguments.

Response: We thank reviewer’s helpful comments. It is widely recognized that the UCL intensity has a power-law of index n with respect to the excitation power, where n effectively represents the number of photons involved in upconversion luminescence (Pollnau M, et al. *Phys. Rev. B* **61**, 3337-3346 (2000)). Both inside-out and local energy transfer architectures, $\text{Yb@Lu}_{0.8}\text{Er}_{0.2}\text{@Lu}$ and $\text{Yb}_{0.5}\text{Lu}_{0.4}\text{Er}_{0.1}\text{@Lu}$, exhibited a typical two-photon upconversion ($n\sim 2.0$). However, for the outside-in $\text{Lu}_{0.8}\text{Er}_{0.2}\text{@Yb@Lu}$, it showed an enhanced UCL with a shallower slope ($n\sim 1.7$) in the power dependence curve. This could be attributed to the relatively efficient energy transfer process in the outside-in structure (including energy migration between Yb^{3+} ions and energy transfer from Yb^{3+} to Er^{3+}) when compared to the inside-out, and the segregation of Er^{3+} and Yb^{3+} that reduces back energy transfer, both of which lead to an enriched Er^{3+} excited intermediates in comparison to the inside-out $\text{Yb@Lu}_{0.8}\text{Er}_{0.2}\text{@Lu}$ and local energy transfer $\text{Yb}_{0.5}\text{Lu}_{0.4}\text{Er}_{0.1}\text{@Lu}$. Therefore, the outside-in UCNPs is less sensitive to the power density change than the other two structures.

Therefore, in the revised manuscript at line 160, we made a change of text from

“The slope of local $\text{Yb}_{0.5}\text{Lu}_{0.4}\text{Er}_{0.1}\text{@Lu}$ UCL intensity-versus-power curve was steeper than that of outside-in $\text{Lu}_{0.8}\text{Er}_{0.2}\text{@Yb@Lu}$ UCNPs. This could be explained by postulating that a higher photon current activated more Yb^{3+} ions, suppressing back energy transfer from Er^{3+} to Yb^{3+} . Therefore, local $\text{Yb}_{0.5}\text{Lu}_{0.4}\text{Er}_{0.1}\text{@Lu}$ still performed poorer, but gradually caught up with outside-in $\text{Lu}_{0.8}\text{Er}_{0.2}\text{@Yb@Lu}$ under stronger irradiance. Unexpectedly, inside-out $\text{Yb@Lu}_{0.8}\text{Er}_{0.2}\text{@Lu}$ had a much dimmer luminescence compared with either outside-in $\text{Lu}_{0.8}\text{Er}_{0.2}\text{@Yb@Lu}$ or local $\text{Yb}_{0.5}\text{Lu}_{0.4}\text{Er}_{0.1}\text{@Lu}$, even though they had similar architectures and compositions of ions, the principal difference being distinct topological arrangements of the sensitizers and activators, with opposite energy migration directions.”

to

“Unexpectedly, inside-out $\text{Yb@Lu}_{0.8}\text{Er}_{0.2}\text{@Lu}$ had a much dimmer luminescence compared with either outside-in $\text{Lu}_{0.8}\text{Er}_{0.2}\text{@Yb@Lu}$ or local $\text{Yb}_{0.5}\text{Lu}_{0.4}\text{Er}_{0.1}\text{@Lu}$, even though they had similar architectures and compositions of ions, the principal difference being distinct topological arrangements of the sensitizers and activators, with opposite energy migration directions. It is widely recognized that the UCL intensity has a power-law of index n with respect to the excitation power, where n effectively represents the number of photons involved in upconversion luminescence⁴⁸. Both inside-out and local energy transfer architectures exhibited a typical two-photon upconversion ($n\sim 2.0$). However, for the outside-in $\text{Lu}_{0.8}\text{Er}_{0.2}\text{@Yb@Lu}$, it showed an enhanced UCL with a shallower slope ($n\sim 1.7$) in the power dependence curve. This could be attributed to the relatively efficient energy transfer process in the outside-in structure (including energy migration between Yb^{3+} ions and energy transfer from Yb^{3+} to Er^{3+}) when compared to the inside-out, and the segregation of Er^{3+} and Yb^{3+} that reduces back energy transfer, both of which lead to an enriched Er^{3+} excited intermediates in comparison to the inside-

out Yb@Lu_{0.8}Er_{0.2}@Lu and local energy transfer Yb_{0.5}Lu_{0.4}Er_{0.1}@Lu. Therefore, the outside-in UCNPs is less sensitive to the power density change than the other two structures.”

5. *Ion-Ion energy transfer simulation methods considering all the possible acceptors in the vicinity of a particular donor were also reported in other works, e.g., 10.1016/j.jlumin.2015.07.005, 10.1021/acs.jpcclett.0c03613, 10.1021/acsnanoscienceau.1c00033, just top mention a few examples. The Monte Carlo method reported here must be commented relatively to those references.*

Response: Thanks for the reviewer’s helpful suggestion. Ion-Ion energy transfer is a typical physical kinetic process. Generally, there are two formalisms for mathematically describing the time behavior for a spatially homogeneous system: the deterministic approach regards the time as a continuous process that is predictable commonly modelled with ordinary differential equations (Gillespie DT. et al. *J. Phys. Chem.* **81**, 2340-2361 (1977); Malta OL. et al. *J. Non Cryst Solids* **354**, 4770-4776 (2008); Shyichuk A, et al. *J. Lumin.* **170**, 560-570 (2016); Neto AC, et al. *J. Lumin.* **210**, 342-347 (2019)); the stochastic approach regards the time evolution as a kind of random walk, which is equivalent to constructing the differential rate equation. In fact, the stochastic approach has a more natural physical basis than the deterministic approach, while the exact expression and solution of the latter are usually mathematically intractable. Using the Monte Carlo procedure to numerically simulate the time evolution correctly accounts for the inherent fluctuations that are necessarily ignored in the deterministic formulation, the feasibility and utility of the simulation algorithm are demonstrated by applying it to the energy transfer as discussed (Qin X, et al. *J. Phys. Chem. Lett.* **12**, 1520-1541 (2021); Mangnus MJJ, et al. *ACS Nanosci. Au* **2**, 111-118 (2022)). Moreover, Monte Carlo method has an edge in dealing with heterogenous systems and is typically spatial aware. Our simplified Monte Carlo approach was inspired by the work of Chen et al (Chen, X et al *Nature Commun.* **7**, 10304 (2016)) and considers only the closest acceptor ion in the short range (<4 Å) of a particular donor. Exchange mechanism exponentially decays with distance ($\propto \exp(-2R/L)$), which is a faster decay compared to other Förster type mechanisms ($\propto R^{-n}$, $n = 6, 8, 10$). Hence, it would be relatively safer to ignore those acceptors in the vicinity but outside the designated short-range (<4 Å) of a particular donor when exchange mechanism is considered dominant.

In brief, better numerical methods combining the stochastic approach and deterministic analysis should be developed to give a better tool to understand the inner works of energy migration. Please refer to the next point for comments added to the manuscript.

6. *The part of the energy migration calculations based on Dexter-type theory together with Monte Carlo simulations describes how many hops the energy in each system (inside-out or outside-in) migrates to the interface of the core-shell structure. The authors used Dexter's theory correctly since the energy transfer between lanthanide*

ions at a short-range distance ($< 4 \text{ \AA}$) has the exchange mechanism (proportional to the electron densities overlap of the Yb-Yb pair) as a dominant one, not considering (correctly) long-range distances once it leads to very low energy transfer rates (see for example Table 2 for distance order from 5 (6.08 \AA) to 20 (9.56 \AA) in 10.1021/acs.jpcllett.0c03613). The energy transfer rates calculations are indeed not easy to evaluate, however, a discussion on the mechanisms involving Ln-Ln energy transfer will be appreciated by the readers. Thus, a brief commentary on the main physical mechanisms (i.e., dipole-dipole, dipole-quadrupole, quadrupole-quadrupole, magnetic dipole-magnetic dipole, and exchange) should be included in the manuscript (see references 10.1021/acs.jpcllett.0c03613).

Response: We thank for the reviewer’s suggestion. The process of energy migration includes exchange mechanism and Förster type energy transfer. And the Förster type energy transfer mechanism including dipole-dipole (W_{d-d}), dipole-quadrupole (W_{d-q}), quadrupole-quadrupole (W_{q-q}) and dipole-magnetic (W_{md-md}) interactions with their rates all inversely proportional to distances to the power of n ($n = 6, 8, 10, 6$). Combining all the energy migration exchange mechanisms, the average energy-transfer rates can be expressed as

$$\langle W \rangle = \sum_i C_i(R_i)W_i(R_i)$$

where R_i is the distance between donor and acceptor, and C_i is the average occurrence number of donors per acceptor ion¹⁶. Besides, the average transfer rate strongly depends on donor-acceptor distance. According to the reported results, long-range distance ($> 4 \text{ \AA}$) average energy transfer rate is at least three orders of magnitude lower than short-range distance ($< 4 \text{ \AA}$) (Qin X, et al. *J. Phys. Chem. Lett.* **12**, 1520-1541 (2021)). Hence, considering that the role of the exchange mechanism is dominant at short distances and other perturbations have small effect at long distances, our numerical approach is justified.

In the revised Manuscript, line 253, we added commentary addressing the two points raised above, as follows:

“Our simulation deals only with the short-range Dexter’s exchange mechanism⁶⁰ when energy migrates along the connected lattice network⁶¹. Förster type multipole interactions do contribute to energy transfer when considering all possible acceptors in the vicinity of a particular donor⁶²⁻⁶⁷. The average transfer rate strongly depends on donor-acceptor distance. Considering that the exchange mechanism is dominant at short distances and other perturbations have small effect at long distances, our numerical approach is justified. More sophisticated spatial aware numerical methods combining the stochastic approach and deterministic analysis should be developed in the future to get a refined picture of how energy migrates in such a delicate system.”

7. The model is very simple with some obvious assumptions. For instance, the pathway of energy migration involving the Yb³⁺ ions located in the outside-in architecture is shorter than that of the inside-out nanoparticles because similar volumes for the core

and the shell were considered (inducing, then, a small shell thickness).

Response: Thanks for the reviewer's comment. It looked intuitively simple but deserves more solid pieces of evidence for us to comprehend the underlying mechanism. Before the model simulation, we first analyzed the thicknesses of the core and shell with back-of-the-envelope calculation as shown in figure 4a, and indeed, the R_c was larger than R_s under the same volume. However, knowing only the radii of different layers, it can be less prudent to claim a faster energy transfer from the shell to the interface without objective proof. The Monte Carlo model helps to visualize how energy migrates through the crystal lattice network in a conditional random walk where the directional probability of migration is governed by Dexter's theory as depicted in Figure 4b. The simulation provided us with a more quantitative answer to how energy migration is taking fewer steps in the shell rather than in the core. The migration pattern also disclosed an elevation dependent heterogeneity of migration steps which reshaped our intuition and makes the explanation of observed differences in outside-in and inside-out architecture more reliable.

Reviewer #3 (Remarks to the Author):

The manuscript by Qian Liu and coworkers addresses an important question regarding the effect of geometric arrangement on energy migration over UCNP sublattices. There were sporadic hints in other literatures that there might be some effect when tweaking contents in core-shell(s) of inorganic nanocrystals which would lead to different optical properties. However, this was the first systematic and quantitative study I have seen to date focusing on the topological effect of swapping core-shell contents of equal volumes. The study was a refined one that their synthesis was able to precisely control equal split of core-shell volumes in such small nanoparticles and the major characterization was done at the single-particle level. The result of the study was also exciting that they provided solid evidences to show that the energy migration was more efficient when going inward from shell to core than an outward one going from core to shell which reminds me of an interesting analogy that detonation of atomic bombs also favors an implosive design. The clearly demonstrated topological effect has a deep philosophical root which hopefully would be appreciated by not only scientists in related fields but also the general public. I would suggest a minor revision before publication.

We thank the reviewer for the positive comments.

Here's the list of the issues that need to be addressed before publication.

1. The color scheme in Fig 1a is a bit confusing. I can see that when orangish Yb ions and blue Er ions were mixed in "local" design, the color they chose for text labeling were green which makes sense. However, in the cartoon the ions were identifiable spheres which in my opinion should not switch to different colors or colors with very strange tones. I would suggest set the ion spheres in the "local" design in same colors as the ones in "outside-in" and "inside-out" designs;

Response: Thanks for the reviewer's helpful suggestion. We have deleted the background of the "local" design and revealed the same color spheres.

2. In Fig. 1g, the outside-in UCNPs showed more than one order of magnitude enhancement than that of inside-outs. Why is there only 6-fold enhancement in the single-particle measurements?

Response: We gratefully appreciate your valuable comment. In Fig.1g, the ensemble assessment of UCNPs was carried out at a power density much lower than the ones used in single particle measurements. The slope differences in the power dependence for the two UCNPs would predict a higher enhancement at lower power density. In addition, we cannot exclude the interference from the discrepancy of nanoparticles' number and the existence of potential aggregation in ensemble measurement. To avoid this interference, more precise upconversion intensity characterization was performed by single-particle imaging. Therefore, we think the single-particle imaging results does not contradict the ensemble measurement. In fact, they were perfectly complementary.

3. In Fig.1h and Supplementary Fig. 2d, the lifetime of inside-out and local architecture UCNPs were similar, but their intensity showed great differences. The authors need to explain this behavior.

Response: Thank you so much for your valuable comment. It is known that the lifetime and upconversion luminescence intensity of UCNPs are not necessarily on the same page (Schroter A, et al. *Adv. Funct. Mater.* **32**, 2113065 (2022)). Although there seems to be a positive correlation in some studies in the past ten years (Tessitore G, et al. *Angew. Chem. Int. Ed. Engl.* **58**, 9742-9751 (2019); Tessitore G, et al. *Adv. Mater.* **32**, e2002266 (2020); Liu X, et al. *Angew. Chem. Int. Ed. Engl.* **60**, 7041-7045 (2021); Zhang Y, et al. *Nat. Commun.* **12**, 6178 (2021)). Here the inside-out architecture has much more migration steps to the interface of the core-shell structure, so the population density and fill rate of Er³⁺ ions' excited state is relatively low, hence the decay rate is relatively fast. The local energy transfer architecture has a high energy migration rate and excited state fill rate but exists nonnegligible back energy transfer, which also accelerates the decay rate. The decay rates of these two architectures are both below the rate for outside-in architecture and influenced by these different factors, they were just close coincidentally.

4. In Fig 3a, the colocalization pattern looks like well matched. If I look closer at the numbers of distances, I can see they aren't identical. Can the authors provide sound justifications in the number discrepancies?

Response: Thanks to the reviewer's helpful comments. In Fig 3a, the distances were determined by the fitted centroids of single-particle luminescence profiles. We used 2D Gaussian fit for localization

$$I(x, y) = \frac{I_0}{2\pi\sigma^2} e^{-\frac{(x-x_0)^2+(y-y_0)^2}{2\sigma^2}} + C$$

where x_0, y_0 are the center coordinates of the point spread function (PSF), σ is the standard deviation of the Gaussian fitting. Although in theory the nanoparticle profile

in SEM is somewhat top-hat shaped, considering the small particle size and its symmetry we could still use similar Gaussian fitting to pinpoint the location of each particle and calculated the distances between nanoparticle pairs. To quantitatively analyze how the positions and/or distances in the wide field luminescence image match those in the SEM image, we did 1st degree 2D polynomial mapping (Pertsinidis A, et al. *Nature* **466**, 647-651 (2010)).

²⁴ using the following equations.

$$\hat{x}_{WF} = p_{00} + p_{01}x_{SEM} + p_{10}y_{SEM} + p_{11}x_{SEM}y_{SEM}$$

$$\hat{y}_{WF} = q_{00} + q_{01}x_{SEM} + q_{10}y_{SEM} + q_{11}x_{SEM}y_{SEM}$$

All 8 nanoparticles' fitted positions in the wide field image and corresponding SEM image were used to generate the mapping coefficient matrixes P, Q .

$$P = \begin{pmatrix} p_{00} & p_{10} \\ p_{01} & p_{11} \end{pmatrix} = \begin{pmatrix} 447.39450 & -0.19931231 \\ 0.90278465 & 8.8982277e-005 \end{pmatrix}$$

$$Q = \begin{pmatrix} q_{00} & q_{10} \\ q_{01} & q_{11} \end{pmatrix} = \begin{pmatrix} -187.35820 & 1.0277292 \\ 0.096447140 & -1.3890537e-005 \end{pmatrix}$$

Figure R13. Overlaid SEM and wide field images with N (N=8) numbered nanoparticles for 2D polynomial mapping.

The registration error for each nanoparticle can be estimated by subtracting registered position from the fitted position in the referencing wide field image.

$$\delta_{reg} = \sqrt{(\hat{x}_{WF} - x_{WF})^2 + (\hat{y}_{WF} - y_{WF})^2}$$

Nanoparticles #1~#5 exhibit registration errors ranging from 7.5 to 19 nm with a median of 9.8 nm, which is ~1/30 of the FWHM of the optical point spread function. Nanoparticle #6, #7 and #8 possess registration errors of 37, 42 and 25 nm respectively, which were ~1/10 of the FWHM, not bad considering their overlapping PSFs. An averaged registration error of ~24 nm for all 8 nanoparticles investigated can be estimated by the following equation.

$$\bar{\delta}_{reg} = \sqrt{\frac{\sum_{i=1}^N [(\hat{x}_{WF,i} - x_{WF,i})^2 + (\hat{y}_{WF,i} - y_{WF,i})^2]}{N}}$$

To summarize, the positional discrepancies between the fluorescence image and SEM micrograph is estimated to be as small as half the size of a nanoparticle (~8 nm) which is well within expected experimental errors and an impressive one considering the registration error is merely a fraction of the size of a nanoparticle.

We made a change of description of the revised Manuscript at line 195 from “we demonstrated that the signal in UCL images was from individual particles, not clusters.”

to

“we did 1st degree 2D polynomial mapping⁵⁶ (see details in Supplementary Note 2) and showed a near perfect match between nanoparticles in the wide field luminescence image and those in the SEM image with registration errors as small as half the size of a single nanoparticle, which demonstrated that the signals in UCL images were from individual particles, not clusters.”

Figure R13 and the related descriptions were added in the revised Supplementary **Note 2**.

5. In Fig 3d, units “pps”, “pps px-1” were not explained anywhere in the text or caption;

Response: Thanks for the reviewer’s questions. In Fig 3d in the original manuscript, pps means photons per second (obtained from the integrated volume of the 2D Gaussian fit), and pps px⁻¹ means photons per second per pixel. We have updated the Fig 3 caption accordingly.

6. In Figure 6, the authors only synthesized the outside-in UCNPs with Er³⁺ doping concentration of 5%. How about the other two architectures and their structure-properties relationship?

Response: Thanks for the reviewer’s helpful suggestion. We synthesized the other two architectures of UCNPs with Er³⁺ doping concentration of 5%, inside-out and local. Similar to the higher Er³⁺ doping, outside-in UCNPs showed the brightest upconversion luminescence. The corresponding results are shown below.

Figure R14. TEM images of UCNP architectures with 5% Er^{3+} doping concentration. TEM images of core (left), core-interior shell (center) and final core-interior shell-inert shell UCNP architectures (right) in local (top), outside-in (middle) and inside-out (bottom) architectures. Each panel includes a size distribution histogram with Gaussian fitting curve; the mean size (by Gaussian fitting) and standard deviation are shown in the lower left corner of each panel.

Figure R15. Luminescence decay curves of UCNP architectures with 5% Er^{3+} doping concentration, excited by 980 nm pulsed laser and recorded at (a) 541 nm; (b) 654 nm emission.

Figure R16. Characterization of UCNPs with 5% Er³⁺ doping concentration. (a) UCL spectra of ensemble UCNPs in cyclohexane solution under 14.6 W/cm² 980 nm laser excitation. (b) UCL spectra of ensemble UCNPs in cyclohexane solution under 530 W/cm² 980 nm laser excitation. (c-d) Power-dependence of green emission at 541 nm and red emission at 654 nm. (e) Saturation curves of single particle brightness at power densities from 126 W/cm² to 21.7 kW/cm² obtained with wide-field microscopy.

In the revised manuscript at line 324, we made the change as following:

From

“In order to find the optimal Er³⁺ doping concentration for upconversion luminescence, the same three nanoparticle architectures studied above were made with 10% or 30% Er³⁺ doping instead of the original 20%. TEM and XRD results demonstrated that these

nanoparticles were uniform in size and were in hexagonal phase (Supplementary Figure 5-8). Single-particle imaging results showed that outside-in architectures with 10% and 30% doping had 4.5 ± 0.2 - and 4.7 ± 0.2 -fold enhancement of luminescence under lower irradiance (680 W/cm^2) when compared with the corresponding inside-out architectures (Figure 6a-b).”

to

“In order to find the optimal Er^{3+} doping concentration for upconversion luminescence, the same three nanoparticle architectures studied above were made with 5%, 10% or 30% Er^{3+} doping instead of the original 20% (Supplementary Figure 5-13). TEM and XRD results demonstrated that these nanoparticles were uniform in size and were in hexagonal phase (Supplementary Figure 5, 7, 9, 10). Single-particle imaging results showed that outside-in architectures with 5%, 10% and 30% doping had 2.8 ± 0.1 -, 4.5 ± 0.2 - and 4.7 ± 0.2 -fold enhancement of luminescence under lower irradiance (680 W/cm^2) when compared with the corresponding inside-out architectures (Figure 6a-b, and Supplementary Figure 12).”

We also added **Figures R14-16** to the Supplementary as **Figures 9-11**.

REVIEWERS' COMMENTS

Reviewer #1 (Remarks to the Author):

The authors have thoroughly addressed the Reviewers' comments. This paper is recommended to be accepted at this time.

Reviewer #2 (Remarks to the Author):

The revised version of the manuscript has been substantially improved. The authors' responses to the questions raised by the referees and the additional work carried out convinced me of the importance of the manuscript that I recommend for publication. I just have one major concern regarding the quantum emission yield (QY) data. The error in QY values is typically 10% and should be represented with 1 significant figure. This corresponds to representing these errors with 1 decimal place and not two, as presented by the authors. Moreover, the proposed explanation for the distinct slopes of the intensity-versus-power curve of Lu_{0.8}Er_{0.2}@Yb@Lu UCNP still to be speculative lacking solid arguments.

Reviewer #3 (Remarks to the Author):

All my concerns were well addressed, so I would like to recommend acceptance of this paper.

Response to reviewer's comments

Reviewer #2 (Remarks to the Author):

The revised version of the manuscript has been substantially improved. The authors' responses to the questions raised by the referees and the additional work carried out convinced me of the importance of the manuscript that I recommend for publication. I just have one major concern regarding the quantum emission yield (QY) data. The error in QY values is typically 10% and should be represented with 1 significant figure. This corresponds to representing these errors with 1 decimal place and not two, as presented by the authors. Moreover, the proposed explanation for the distinct slopes of the intensity-versus-power curve of Lu_{0.8}Er_{0.2}@Yb@Lu UCNP still to be speculative lacking solid arguments.

1. The error in QY values is typically 10% and should be represented with 1 significant figure. This corresponds to representing these errors with 1 decimal place and not two, as presented by the authors.

Response: Thanks for the reviewer's helpful suggestion. We revised the UCQY results as Supplementary Table 4.

Supplementary Table 4. The summary of UCQY for Lu-based samples.

Nanoparticle	Type	UCQY
Lu _{0.95} Er _{0.05} @Yb@Lu	Outside-in	1.6±0.4%
Yb@Lu _{0.95} Er _{0.05} @Lu	Inside-out	0.4±0.2%
Yb _{0.5} Lu _{0.475} Er _{0.025} @Lu	Local energy transfer	0.8±0.4%
Lu _{0.9} Er _{0.1} @Yb@Lu	Outside-in	1.9±0.4%
Yb@Lu _{0.9} Er _{0.1} @Lu	Inside-out	0.4±0.2%
Yb _{0.5} Lu _{0.45} Er _{0.05} @Lu	Local energy transfer	0.7±0.3%
Lu _{0.8} Er _{0.2} @Yb@Lu	Outside-in	1.6±0.5%
Yb@Lu _{0.8} Er _{0.2} @Lu	Inside-out	0.4±0.2%
Yb _{0.5} Lu _{0.4} Er _{0.1} @Lu	Local energy transfer	0.6±0.2%
Lu _{0.8} Er _{0.2} @Yb _{0.75} Lu _{0.25} @Lu	Outside-in, interior shell	1.5±0.3%
Lu _{0.8} Er _{0.2} @Yb _{0.5} Lu _{0.5} @Lu	Outside-in, interior shell	1.3±0.3%
Lu _{0.7} Er _{0.3} @Yb@Lu	Outside-in	1.2±0.4%
Yb@Lu _{0.7} Er _{0.3} @Lu	Inside-out	0.3±0.1%
Yb _{0.5} Lu _{0.35} Er _{0.15} @Lu	Local energy transfer	0.5±0.2%
Yb _{0.2} Lu _{0.78} Er _{0.02} @Lu	traditional	0.7±0.4%

2. The proposed explanation for the distinct slopes of the intensity-versus-power curve of $\text{Lu}_{0.8}\text{Er}_{0.2}@Yb@Lu$ UCNP still to be speculative lacking solid arguments.

Response: We thank reviewer's helpful comments. We revised the description about the intensity-versus-power curve of $\text{Lu}_{0.8}\text{Er}_{0.2}@Yb@Lu$ UCNP to a hypothesize, as follows:

“Both inside-out and local energy transfer architectures exhibited a typical two-photon upconversion ($n\sim 2.0$). However, for the outside-in $\text{Lu}_{0.8}\text{Er}_{0.2}@Yb@Lu$, it showed an enhanced UCL with a shallower slope ($n\sim 1.7$) in the power dependence curve. We speculated that there was a relatively efficient energy transfer process in the outside-in structure when compared to the inside-out, and the segregation of Er^{3+} and Yb^{3+} that reduces back energy transfer, both of which lead to an enriched Er^{3+} excited intermediates in comparison to the inside-out $\text{Yb}@Lu_{0.8}\text{Er}_{0.2}@Lu$ and local energy transfer $\text{Yb}_{0.5}\text{Lu}_{0.4}\text{Er}_{0.1}@Lu$. Therefore, the outside-in UCNPs is less sensitive to the power density change than the other two structures.”